# Signal Propagation in Transformers: Theoretical Perspectives and the Role of Rank Collapse

**Lorenzo Noci**[*1]
lorenzo.noci@inf.ethz.ch

**Sotiris Anagnostidis**[*1]
sotirios.anagnostidis@inf.ethz.ch

**Luca Biggio**[*1,2]
luca.biggio@inf.ethz.ch

**Antonio Orvieto**[*1]
antonio.orvieto@inf.ethz.ch

**Sidak Pal Singh**[*1,3]
sidak.singh@inf.ethz.ch

**Aurelien Lucchi**[4]
aurelien.lucchi@unibas.ch

## Abstract

Transformers have achieved remarkable success in several domains, ranging from natural language processing to computer vision. Nevertheless, it has been recently shown that stacking self-attention layers — the distinctive architectural component of Transformers — can result in rank collapse of the tokens' representations at initialization. The question of if and how rank collapse affects training is still largely unanswered, and its investigation is necessary for a more comprehensive understanding of this architecture. In this work, we shed new light on the causes and the effects of this phenomenon. First, we show that rank collapse of the tokens' representations hinders training by causing the gradients of the queries and keys to vanish at initialization. Furthermore, we provide a thorough description of the origin of rank collapse and discuss how to prevent it via an appropriate depth-dependent scaling of the residual branches. Finally, our analysis unveils that specific architectural hyperparameters affect the gradients of queries and values differently, leading to disproportionate gradient norms. This suggests an explanation for the widespread use of adaptive methods for Transformers' optimization.

## 1 Introduction

Since its first appearance in Vaswani et al. [2017], the Transformer architecture has revolutionized the field of Natural Language Processing (NLP), achieving remarkable success in tasks such as text classification [Yang et al., 2019], machine translation [Conneau and Lample, 2019], reading comprehension [Brown et al., 2020] and question answering [Raffel et al., 2019] among others. Recent efforts have effectively extended its applicability to computer vision [Dosovitskiy et al., 2020] and other domains [Baevski et al., 2020, Huang et al., 2018, Biggio et al., 2021, Polu et al., 2022], further popularizing it outside NLP.

The Transformer operates on inputs comprising a sequence of tokens. At its core, it relies on stacked attention layers, which compute a measure of relevance for the whole sequence by assigning token-wise importance weights — obtained by matrix multiplication of the *queries* and *keys*, and finally normalized with the softmax function. The output of an attention layer is then a linear combination

---

[1]Dept of Computer Science, ETH Zürich, [2]Robotics & ML, CSEM SA, Alpnach, Switzerland, [3]MPI for Intelligent Systems, Tübingen, [4]Department of Mathematics and Computer Science, University of Basel

36th Conference on Neural Information Processing Systems (NeurIPS 2022).

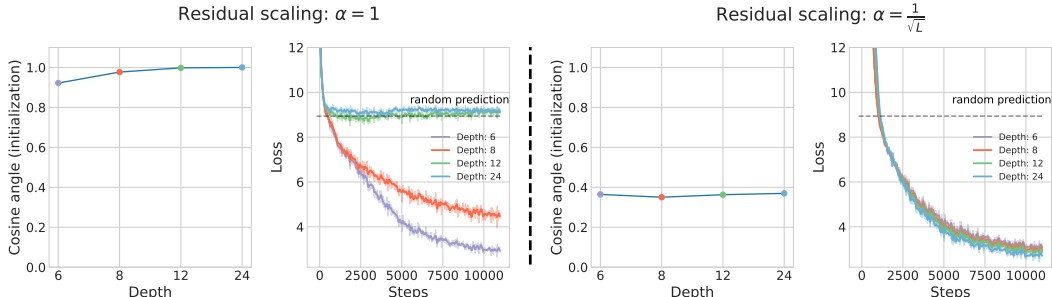

Figure 1: Evolution of the cosine of the angle between tokens for training POST-LN Transformers of increasing depth, with the Adam optimizer, for the IWSLT'14 De-En translation task. Unless adequate residual scaling is used at initialization, increasing depth leads to an increase in the tokens' alignment at initialization, which can inhibit training.

of the importance weights and the so-called *values*. Then, the architecture includes fully-connected sub-layers, residual connections [He et al., 2016], and layer normalization (LN), as illustrated in Fig. 2.

In the absence of residual connections, Dong et al. [2021] proved that at initialization the rank of the sequence representation collapses doubly exponentially with depth, and both layer normalization and fully connected layers can only partially alleviate the speed of degeneracy. Under *rank collapse*, the model does not distinguish between representations of different tokens, which are perfectly aligned in feature space at initialization. However, the precise implications of rank collapse in Transformers are not fully understood.

In this paper, we show that a high alignment of the tokens' representations at initialization — corresponding to rank collapse in the extreme case of perfect alignment — affects training by causing vanishingly small gradients of the queries and keys' parameter matrices. This problem severely diminishes the capabilities of the model to learn meaningful attention weights and is further exacerbated in very deep networks, where the rank deficiency — and hence the vanishing gradient problem of the queries and keys — affects several layers (see Fig. 1). In order to shed light on this problem, we take inspiration from the flourishing literature on signal propagation in random networks and start our analysis by computing the expected gradients of an attention layer with respect to the queries, keys, and values, which leads to Theorem 3.2 on the vanishing gradients for the queries and keys. From here, we pursue two different directions.

Firstly, we investigate under which conditions rank collapse can be avoided by studying the evolution of the input sequence in a Transformer at initialization. Our theory reveals that a depth-dependent scaling of the residual branches, beyond stabilizing the norm of the activations at initialization, also approximately preserves the cosine of the angle between tokens, hence stabilizing the rank of the propagating sequence. We show that this holds even in the infinite-depth limit.

Secondly, we illustrate that there are factors, other than the average tokens' correlation, that affect differently the gradient norm of the queries and keys compared to the values. In particular, the propagating sequence's squared norm has a linear dependence in the values, while a cubic one in the queries and keys, justifying the use of layer normalization. We also highlight a different dependence on the embedding dimension and the length of the input sequence, implying that the gradient norm of a subset of parameters can potentially be of different orders of magnitude, as empirically hinted by previous works [Liu et al., 2020]. Our analysis brings to light fundamental issues in the signal propagation in Transformers, opening the way for new, well-founded and motivated approaches to improve optimization in these models.

## 2 Background

**Transformers.** A Transformer architecture consists of $L$ stacked attention blocks, as show in Fig. 2. Layer normalization is usually applied token-wise either after the residual connections or to the inputs

of the self-attention and position-wise feed-forward sub-layers, leading to the POST-LN [Vaswani et al., 2017] and PRE-LN [Wang et al., 2019, Xiong et al., 2020] variants respectively.

Formally, given an input sequence $\mathbf{X} \in \mathbb{R}^{n \times d_v}$, with $n$ tokens of dimension $d_v$, the single-head unmasked scaled dot-product self-attention[1] is defined as:

$$\mathbf{S}^\ell := \mathbf{A}^\ell \mathbf{X}^\ell \mathbf{W}^V, \text{ where } \mathbf{A}^\ell = \text{softmax}\left(\frac{1}{\sqrt{d_k}} \mathbf{X}^\ell \mathbf{W}^Q \left(\mathbf{X}^\ell \mathbf{W}^K\right)^\top\right), \tag{1}$$

where the softmax function is applied independently across each row, and the superscript $\ell$ indexes the $\ell$-th layer. The matrices $\mathbf{W}^Q, \mathbf{W}^K \in \mathbb{R}^{d_v \times d_k}$ and $\mathbf{W}^V \in \mathbb{R}^{d_v \times d_v}$ are learnable parameters, and each layer is initialized with an independent set of weights. In the literature, the matrices $\mathbf{X}^\ell \mathbf{W}^Q, \mathbf{X}^\ell \mathbf{W}^K, \mathbf{X}^\ell \mathbf{W}^V$ are referred to as queries, keys and values, respectively. The complete Transformer block, in the absence of layer normalization, can be written recursively as:

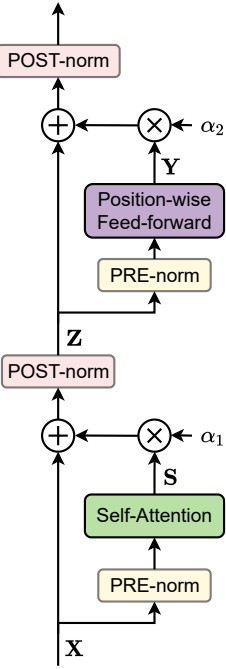

$$\mathbf{Z}^\ell = \alpha_1 \mathbf{S}^\ell + \mathbf{X}^\ell \tag{2}$$
$$\mathbf{Y}^\ell = \sigma(\mathbf{Z}^\ell \mathbf{W}^{F_1}) \mathbf{W}^{F_2} \tag{3}$$
$$\mathbf{X}^{\ell+1} = \alpha_2 \mathbf{Y}^\ell + \mathbf{Z}^\ell, \tag{4}$$

where the introduced $\alpha_1, \alpha_2$ parameters indicate the strength of the residual block, $\mathbf{W}^{F_1}, \mathbf{W}^{F_2} \in \mathbb{R}^{d_v \times d_v}$ [2] are matrices of learnable parameters; we set $\mathbf{X}^0 := \mathbf{X}$, and $\sigma : \mathbb{R} \to \mathbb{R}$ is an activation function. In our case, $\sigma$ is the ReLU function, but we relax this assumption to the linear activation from Section 3.2 on.

At initialization, each weight is sampled independently from a distribution with zero-mean and variance $\sigma_v^2 = \frac{1}{d_v}$ for the values and feedforward weights[3], and $\sigma_k^2 = \frac{1}{d_k}$ for the queries and keys. This is the standard "Xavier" [Glorot and Bengio, 2010] or "He" [He et al., 2015] initialization, commonly used in deep learning.

Figure 2: A single Transformer block.

**Rank Collapse in Transformers.** Interestingly, Dong et al. [2021] proved that when the residual branches are omitted, the matrix of the tokens' representations $\mathbf{X}^\ell$ converges to a rank-1 matrix in which all the representations are the same and equal to a vector $\boldsymbol{x} \in \mathbb{R}^{d_v}$, i.e. $\mathbf{X}^\ell \to \mathbf{1}_n \boldsymbol{x}^\top$, where $\mathbf{1}_{d_v}$ is the vector with all ones in $\mathbb{R}^{d_v}$. Note that this is a slightly stronger notion of a rank-1 matrix, as it implies that all the tokens' representations are both perfectly aligned and have the same norm. Indicating the inner product with the usual bracket notations $\langle \cdot, \cdot \rangle$, and the cosine of the angle between two tokens as $\theta_{k,k'}$, perfect alignment happens when $\langle \mathbf{X}_k^\ell, \mathbf{X}_{k'}^\ell \rangle = \left\|\mathbf{X}_k^\ell\right\| \left\|\mathbf{X}_k^\ell\right\| \cos \theta_{k,k'}$ with $\cos \theta_{k,k'} = 1$ for all $k, k' \in [n]$. Note that perfect alignment together with equal norm between all the tokens implies that all the representations are the same. One of our main contributions is to provide an explanation of how rank collapse affects the gradients of a Transformer at initialization.

**Vanishing Gradient Problem.** Traditionally considered one of the core issues that prevents successful training, the vanishing gradient problem has a long and rich history that dates back to before the popularization of deep learning [Hochreiter, 1991, Bengio et al., 1994]. In its essence, given a loss function $\mathcal{L} : \mathbb{R}^{n \times d_v} \to \mathbb{R}$, vanishing gradients occur when the norm of the gradient of the loss $\mathcal{L}$ with respect to the parameters of the network $\mathbf{W}$ — which we indicate as $\left\|\frac{\partial \mathcal{L}}{\partial \mathbf{W}}\right\|$ — is too small to provide enough backpropagating signal, thus hindering gradient-based optimization methods.

---

[1]Our analysis also easily generalizes to the case of cross-attention.

[2]In practice, one commonly uses $\mathbf{W}^{F_1} \in \mathbb{R}^{d_v \times d_F}, \mathbf{W}^{F_2} \in \mathbb{R}^{d_F \times d_v}$ where $d_F = \gamma d_v$, with $\gamma \in \{2, 4, 8\}$. Our results then hold up to a constant factor that depends on $\gamma$.

[3]One should explicitly write the layer dependence $\mathbf{W}^{Q,\ell}, \mathbf{W}^{K,\ell}, \mathbf{W}^{V,\ell}, \mathbf{W}^{F_1,\ell}, \mathbf{W}^{F_2,\ell}$. We at times suppress the $\ell$ index to improve readability. In case $\sigma$ is the ReLU function, we set $\mathbf{W}^{F,1}$ to have variance $\frac{2}{d_v}$.

Despite extensive research toward understanding and overcoming the problem in disparate contexts [Glorot and Bengio, 2010, He et al., 2015, Hanin, 2018, Zhang et al., 2019], a formal explanation of its role in relatively new architectures such as Transformers is largely missing in the literature, with a few exceptions [Xiong et al., 2020, Wang et al., 2022, Huang et al., 2020]. In our paper (Section 3.1), we show how vanishing gradient occurs in conjunction with the rank collapse issue identified by Dong et al. [2021].

**Signal Propagation in Random Networks at Initialization.** After addressing the question on the effects of rank collapse, we take a step back and rigorously analyze its causes by looking at how the properties of the input sequence $\mathbf{X}$ are lost/preserved as it propagates through a randomly initialized Transformer. More specifically, we focus on two aspects of the propagating sequence: the expected Frobenius norm $\mathbb{E}\left\|\mathbf{X}^\ell\right\|^2$ and the expected inner product between different tokens $\mathbb{E}\langle\mathbf{X}_k, \mathbf{X}'_k\rangle$, with $k \neq k'$. The former is linked to a number of studies on the initialization of neural networks at the *edge of chaos* [Poole et al., 2016, Schoenholz et al., 2017], and vanishing/exploding gradients [Hanin, 2018]. The latter quantity describes how the geometry of the feature space changes after applying a Transformer block, and is related to the concept of *dynamical isometry* [Saxe et al., 2013]. To understand the evolution of the inner product, we analyze the following measure of correlation [Nachum et al., 2021, Cho and Saul, 2009]:

$$\rho^\ell_{kk'} := \frac{\mathbb{E}\langle\mathbf{X}^\ell_k, \mathbf{X}^\ell_{k'}\rangle}{\sqrt{\mathbb{E}\left\|\mathbf{X}^\ell_k\right\|^2 \mathbb{E}\left\|\mathbf{X}^\ell_{k'}\right\|^2}}. \tag{5}$$

Note that $\rho^\ell_{kk'} = 1$ if and only if the $k$-th and $k'$-th tokens are perfectly aligned ($\cos\theta_{kk'} = 1$). We stress that in our case — differently from the aforementioned works — instead of analyzing the relationship between two different data points, we study the relationship between tokens of the same sequence.

## 3 Theoretical Results

The goal of this section is twofold. In Lemma 3.1 and Theorem 3.2, we provide an explanation of the possible cause of vanishingly small gradients for queries and keys at initialization, namely the high correlations between the tokens representations in $\mathbf{X}^\ell$ as the depth increases. In Section 3.2, we show that the problem can be mitigated with an appropriate choice of the residual branch parameters $\alpha_1$ and $\alpha_2$ that inversely scales with the depth of the network $L$. Under the proposed scaling, the correlations are well behaved even in the infinite depth limit. Finally, in Section 3.3 we analyze the scaling of the gradients with respect to other network's parameters, and in 3.4 we draw some connections between our findings and optimization of Transformers.

### 3.1 Vanishing Gradients for Queries and Keys under Rank Collapse

To investigate the problem of vanishing gradients in the attention layers, we make use of the framework of matrix calculus [Magnus and Neudecker, 2019, Singh et al., 2021]. In particular, we compare the expected Frobenius norm of the gradient of a self-attention layer with respect to its parameters $\mathbb{E}\left\|\frac{\partial \mathbf{S}^\ell}{\partial \mathbf{W}}\right\|^2_F$, where here $\mathbf{W}$ indicates one of the keys, queries or values weight matrices. Due to the well-known difficulty of computing expectations of the softmax [Daunizeau, 2017, Shekhovtsov and Flach, 2018], throughout this manuscript, we make the simplifying assumption that the softmax output is the uniform distribution at initialization, i.e. the $n \times n$ matrix containing $\frac{1}{n}$ in each entry.

**Assumption 3.1** (Uniform attention). *We assume that* $\mathbf{A}^\ell = \frac{1}{n}\mathbf{1}_{n \times n}$,

where $\mathbf{1}_{n \times n}$ is the matrix with all entries equal to $1$. Crucially, in Appendix A.5, we formally show that *this assumption holds almost surely* in the limit $d_k \to \infty$. There, we also experimentally show that even in the more realistic case where $d_k = d_v \approx 512$, the empirical simulations provide a surprisingly faithful approximation of the theoretical insights presented in this paper.

We define the mean token $\bar{\mathbf{x}}^\ell$ through its components $\bar{x}^\ell_i = \frac{1}{n}\sum_{k=1}^n \mathbf{X}^\ell_{ki}, i \in [d_v]$. In the following theorem, we compute the expected gradients of an attention layer at initialization, and set the basis for our following analysis. We provide the results only for the queries, as the case for the keys is analogous.

**Lemma 3.1.** *Let $\mathbf{X}^\ell$ be the representations of the input sequence at the $\ell$-th layer. Under the uniform-attention assumption, we have*

$$\mathbb{E}\left\|\frac{\partial \mathbf{S}^\ell}{\partial \mathbf{W}^{V,\ell}}\right\|_F^2 = d_v n \mathbb{E}\|\bar{\boldsymbol{x}}^\ell\|^2 \; ; \tag{6}$$

$$\mathbb{E}\left\|\frac{\partial \mathbf{S}^\ell}{\partial \mathbf{W}^{Q,\ell}}\right\|_F^2 = \frac{\sigma_v^2 \sigma_k^2 d_v}{n^2} \cdot \mathbb{E}\left[\|\mathbf{X}^\ell\|_F^2 \cdot \|(\mathbf{X}^\ell)^\top \mathbf{X}^\ell - n\bar{\boldsymbol{x}}^\ell(\bar{\boldsymbol{x}}^\ell)^\top\|_F^2\right] \; ; \tag{7}$$

$$\mathbb{E}\left\|\frac{\partial \mathbf{S}^\ell}{\partial \mathbf{X}^\ell}\right\|_F^2 \leq \frac{8\sigma_q^2 \sigma_k^2 \sigma_v^2 d_k d_v}{n} \cdot \mathbb{E}\left\|(\mathbf{X}^\ell)^\top \mathbf{X}^\ell - n\bar{\boldsymbol{x}}^\ell(\bar{\boldsymbol{x}}^\ell)^\top\right\|_F^2 + 2d_v^2\sigma_v^2 \; . \tag{8}$$

We defer the precise study of the scaling of these quantities as a function of $n$ and $d_v, d_k$, to Section 3.3. At this stage, it is crucial to note that $\frac{1}{n}(\mathbf{X}^\ell)^\top \mathbf{X}^\ell - \bar{\boldsymbol{x}}^\ell(\bar{\boldsymbol{x}}^\ell)^\top$ is the centered empirical covariance matrix of the tokens' representations. It is easy to see that if $\mathbf{X}^\ell$ is a rank-1 matrix, then all the rows of $\mathbf{X}^\ell$ are proportional to a fixed $d_v$-dimensional vector, and the empirical covariance matrix has all zero entries. Introducing a differentiable loss function $\mathcal{L}: \mathbb{R}^{n \times d_v} \to \mathbb{R}$, we make the statement on vanishing gradients more formal in the following theorem:

**Theorem 3.2** (Vanishing gradients under rank collapse). *Suppose that the uniform-attention assumption holds. If additionally $\mathbf{X}^\ell$ for any $l \in [L]$ has rank-1, and there exists a vector $\boldsymbol{x} \in \mathbb{R}^d$ such that $\mathbf{X}^\ell = \mathbf{1}_n \boldsymbol{x}^T$, then:*

$$\mathbb{E}\left\|\frac{\partial \mathcal{L}}{\partial \mathbf{W}^{Q,\ell}}\right\|_F^2 = 0, \quad \mathbb{E}\left\|\frac{\partial \mathcal{L}}{\partial \mathbf{W}^{K,\ell}}\right\|_F^2 = 0, \tag{9}$$

*where the expectation is taken over the weight matrices. This implies that these quantities are vanishing almost surely, due to the non-negativeness of the norm.*

The proof simply relies on expanding the norm of the gradient of the loss with the aid of the chain rule and then bounding it by the product of the norms of each term of the chain. The final result holds with an application of Lemma 3.1, in which the rank-1 assumption makes $\mathbb{E}\left\|\frac{\partial \mathbf{S}^\ell}{\partial \mathbf{W}^{Q,\ell}}\right\|$ vanish. The proof of the results presented in this section is deferred to Appendix A.1.

In light of Theorem 3.2, we can conclude that the issue of rank collapse originally identified in Dong et al. [2021] corresponds to an initialization in a region of vanishing gradient signal in the subspace of parameters identified by the queries and keys. How can this affect training? One may argue that if rank collapse does not happen in the very first layer, then the corresponding gradients are non-zero and the rank of the subsequent layers — affected by rank collapse — can be increased with the first few steps of gradient descent. In practice, we show empirically in Fig. 1 that escaping this pathological landscape is harder in deeper nets where rank collapse persists across several layers.

## 3.2 Forward Signal Propagation and the Importance of Scaling the Residual Branches

We now turn our attention to the study of the influence of skip connections in Transformers. Dong et al. [2021] showed that simply adding this architectural trick prevents rank collapse. Somewhat surprisingly, we show that while the claim holds for any finite depth, the average angle between different tokens quickly increases with just a few layers, and as $L \to \infty$ a Transformer can still lose rank unless the residual branches are adequately initialized. As Dong et al. [2021] showed that layer normalization does not avoid rank collapse, we omit it in our analysis. Firstly, we introduce two lemmas on the propagation of inner products (Lemma 3.2) and the norm (Lemma 3.3) of the tokens' representations.

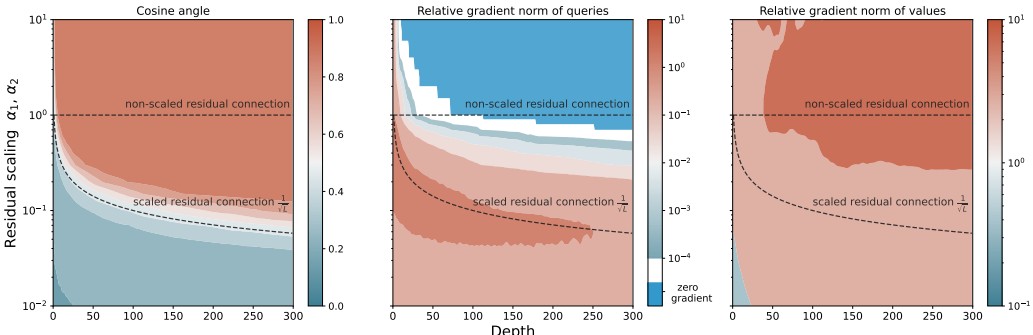

Figure 3: Effect of the residual scaling to the norm of the gradients of the network at initialization with respect to some loss. From left to right: (a) the cosine of the angle between tokens increases with depth. Note how larger values of $\alpha_1, \alpha_2$ imply a faster token alignment with depth (Theorem 3.3). Subplots (b) and (c) show the gradients of the queries-keys and values parameters respectively by increasing depth, compared to the corresponding norms of the first layer. Gradients for the queries-keys diminish with depth, while the opposite happens for the values. We use POST-LN to disentangle the effect of the variance of the input.

**Lemma 3.2** (Propagation of inner products). *Let* $C(\mathbf{X}^\ell) = \sum_{k,k'} \langle \mathbf{X}^\ell_k, \mathbf{X}^\ell_{k'} \rangle$ *and* $\mathbf{X}$ *the input sequence. Under the Assumption 3.1 and if $\sigma$ is the linear activation function, we have that:*

$$\mathbb{E}\left[C(\mathbf{X}^L)\right] = (\alpha_2^2 + 1)^L (\alpha_1^2 + 1)^L C(\mathbf{X}). \tag{10}$$

*hence, under the depth scaling for the residual block parameters $\alpha_1^2 = \frac{\tilde{\alpha}_1}{L}, \alpha_2^2 = \frac{\tilde{\alpha}_2}{L}$ with $\tilde{\alpha}_1, \tilde{\alpha}_2 \in \mathbb{R}$ independent of $L$, we have that:*

$$\lim_{L \to \infty} \mathbb{E}[C(\mathbf{X}^L)] = e^{\tilde{\alpha}_1 + \tilde{\alpha}_2} C(\mathbf{X}). \tag{11}$$

Note that $C(\mathbf{X}^\ell) = n^2 \left\|\bar{\boldsymbol{x}}^\ell\right\|^2$. The lemma on the propagation of the norm is slightly more involved:

**Lemma 3.3** (Propagation of the norm). *Let* $\mathbf{X}^L$ *be the representations of the input sequence at the final layer. Under the assumptions of Lemma 3.2, we have that:*

$$\mathbb{E}\left\|\mathbf{X}^L\right\|_F^2 = n(\alpha_2^2 + 1)^L \alpha_1^2 \sum_{k=0}^{L-1} (\alpha_1^2 + 1)^k \left\|\bar{\boldsymbol{x}}\right\|^2 + (\alpha_2^2 + 1)^L ||\mathbf{X}||_F^2, \tag{12}$$

*hence, under the depth scaling for the residual block parameters $\alpha_1^2 = \frac{\tilde{\alpha}_1}{L}, \alpha_2^2 = \frac{\tilde{\alpha}_2}{L}$ with $\tilde{\alpha}_1, \tilde{\alpha}_2 \in \mathbb{R}$ independent of $L$, we have that:*

$$\lim_{L \to \infty} \mathbb{E}\left\|\mathbf{X}^L\right\|_F^2 = ne^{\tilde{\alpha}_2}(e^{\tilde{\alpha}_1} - 1)\left\|\bar{\boldsymbol{x}}\right\|^2 + e^{\tilde{\alpha}_2} ||\mathbf{X}||_F^2. \tag{13}$$

The proof of Lemma 3.3 consists in expanding $\mathbb{E}\left\|\mathbf{X}^L\right\|_F^2$ according to the defining equations for the Transformer, and simplifying the expression by using iterated expectations $\mathbb{E}\left\|\mathbf{X}^L\right\|_F^2 = \mathbb{E}[\mathbb{E}[\left\|\mathbf{X}^L\right\|_F^2 \mid \mathbf{X}^\ell]]$ to exploit the conditional independence between different layers, and then computing the expectations using the independence assumption on the weights. The expression on the right-hand side will then depend on $\mathbf{X}^\ell$ only through its norm $\left\|\mathbf{X}^\ell\right\|$ and the norm of the mean token $\left\|\bar{\boldsymbol{x}}^\ell\right\|^2$. Using Lemma 3.2 then allows us to unroll the recursion and get the final result. The complete proof, together with the proof of Lemma 3.2, can be found in Appendix A.3.

The previous Lemma provides theoretical justification that scaling the residual branches by setting the alpha parameters to be $\mathcal{O}(1/\sqrt{L})$ allows both the norm of the propagating input and the inner products between different tokens to be approximately preserved. Hence, the information contained in the input is not lost, even in the infinite depth limit.

**Residual Scaling Preserves Correlations.** We now prove that without the depth-dependent residual scaling (i.e. with $\alpha_1 = \alpha_2 = 1$) the correlation between the tokens quickly increases, and reaches perfect alignment in the infinite depth limit. More specifically, our argument shows that in this limit, the correlation between different tokens $\rho^\ell_{k,k'}$ as in Eq. (5) converges to 1, implying rank collapse. Furthermore, we show how setting the residual parameters $\alpha_1$ and $\alpha_2$ as dictated by Theorem 3.3, ensures that the correlation measure is dependent on the input in a non-trivial way even at infinite depth. To this end, we introduce the average correlation at layer $\ell$:

$$\rho^\ell = \frac{1}{n(n-1)} \sum_{k \neq k'} \rho^\ell_{kk'}. \tag{14}$$

Note that $\rho^\ell = 1$ if and only if every pair of tokens is perfectly aligned. We are now ready to formalize the influence of the $1/\sqrt{L}$-scaling on the correlation between tokens' representations by stating Theorem 3.3.

**Theorem 3.3.** *Let the input tokens have the same norm, i.e.* $\|\mathbf{X}_k\| = \|\boldsymbol{x}\| \ \forall k \in [n]$ *for some* $\boldsymbol{x} \in \mathbb{R}^{d_v}$. *Under the depth scaling for the residual block parameters* $\alpha_1^2 = \frac{\tilde{\alpha}_1}{L}, \alpha_2^2 = \frac{\tilde{\alpha}_2}{L}$ *with* $\tilde{\alpha}_1, \tilde{\alpha}_2 \in \mathbb{R}$ *independent of* $L$, *we have that:*

$$\lim_{L \to \infty} \rho^\ell = \frac{n e^{\tilde{\alpha}_1} C(\mathbf{X})}{(n-1)[(e^{\tilde{\alpha}_1} - 1)C(\mathbf{X}) + n \|\mathbf{X}\|_F^2]} - \frac{1}{n-1}. \tag{15}$$

*On the other hand, if* $\alpha_1, \alpha_2 \neq 0$ *are some constants independent of* $L$, *we have that:*

$$\lim_{L \to \infty} \rho^\ell = 1. \tag{16}$$

The proof consists in noting that due to the symmetry of the problem at initialization, for a fixed layer the expected norm of each token is the same. Hence, by our definition of $\rho^\ell_{kk'}$, we can write $\mathbb{E}\langle \mathbf{X}^\ell_k, \mathbf{X}^\ell_{k'} \rangle = \rho^\ell_{kk'} \mathbb{E} \|\boldsymbol{x}^\ell\|^2$. By summing over the $k, k'$ indexes, the resulting equation will depend on $\mathbb{E}[C(\mathbf{X}^\ell)]$ and $\mathbb{E} \|\mathbf{X}^\ell\|^2$, which can be expanded using Lemma 3.2 and 3.3 respectively. The result is then given by solving for $\rho^\ell$.

Note that under the $1/\sqrt{L}$-scaling, the correlation term is one if and only if $C(\mathbf{X}) = n \|\mathbf{X}\|^2$, which holds in the degenerate case where all the input tokens are perfectly aligned. In Appendix A.4, we give precise formulas for the expected correlations at any depth, showing that $\rho^\ell$ reaches values close to one even for relatively shallow networks when the $1/\sqrt{L}$-scaling is not adopted (see also Fig. 3 (left)). Additionally, in Fig. 4, we empirically show that in the presence of the $1/\sqrt{L}$-scaling, layer normalization (either PRE or POST) does not significantly affect the evolution of the correlations. On the other hand, without the residual scaling, PRE-LN seems to alleviate the rate of increase of $\rho^\ell_{kk'}$. It is intriguing that most deep Transformer models use this configuration [Brown et al., 2020]. We provide more extensive empirical results in Appendix B.

Note that the $1/\sqrt{L}$ scaling for the residual branches has been previously studied in the context of stabilization of residual networks (see Section 4), here we extend these results to Transformers and provide new insights on its role in the context of rank preservation. Finally, note that by setting $\tilde{\alpha}_1, \tilde{\alpha}_2 = 0$, we recover the so called "ReZero" initialization [Bachlechner et al., 2021]. In this context, the $1/\sqrt{L}$ scaling extends this framework as it allows for wider range of values for $\tilde{\alpha}_1, \tilde{\alpha}_2$ while still guaranteeing stability.

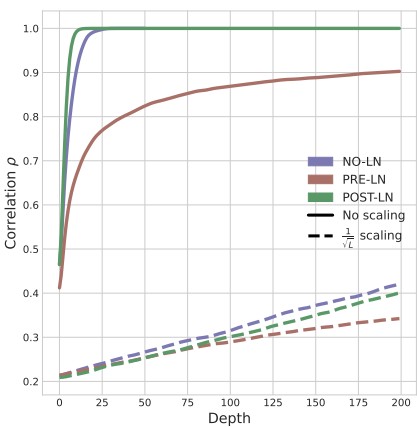

Figure 4: Evolution of Correlation in Transformers with (dashed lines) and without (solid lines) $1/\sqrt{L}$-scaling for PRE-LN, POST-LN and without layer normalization (No-LN).

**ReLU extension.** We mention here that extending these results from the linear activation to the ReLU case is known to be a hard problem, due to the technical difficulty of propagating the inner products across ReLU layers that are shared among the tokens (this is the case in the position-wise feed-forward layers in Transformers). Exact formulas can be found only in the case of one ReLU layer with Gaussian inputs in Cho and Saul [2009]. However, in the context of rank collapse analyzed here, the linear activation function provides a bound on the correlation with respect to the ReLU case. In fact, correlations are exactly preserved in expectation in the linear case, but increase in the ReLU case (for instance, see the contraction argument in Nachum et al. [2021] below Equation (2)). Hence, the perfect alignment (a.k.a rank collapse) that affects the linear case affects the ReLU case as well (in which case the rank collapses even faster with depth, as we show in Figure 10).

## 3.3    Dependence on the Angle between Tokens and the Input Norm

In this section, we drop the superscript $\ell$ as it is obvious from context and assume for simplicity that $d_k = d_v$. To gain a better intuition on the factors that affect the gradients and provide additional insights, we study the case in which every pair of distinct tokens are zero-mean Gaussian random variables, correlated in the same way, i.e $\rho_{ii'}^\ell = \rho$ for $i \neq i'$ or more precisely

$$\mathbb{E}\left[\mathbf{X}_{i,j}\mathbf{X}_{i',j'}\right] = \begin{cases} 0 & j \neq j' \ \text{(independent dimensions)} \\ \sigma_x^2 & i = i', j = j' \\ \rho\sigma_x^2 & i \neq i', j = j' \end{cases}.$$

To see that this equation satisfies our definition of the correlation metric, note that $\mathbb{E}[\|\mathbf{X}_i\|^2] = d\sigma_x^2$ and $\mathbb{E}\langle\mathbf{X}_i, \mathbf{X}_{i'}\rangle = d\sigma_x^2\rho$, for $i \neq i'$. Then, the expected norm of the gradients for the values (Eq. (6)) simplifies to

$$\mathbb{E}\left\|\frac{\partial\mathbf{S}}{\partial\mathbf{W}^V}\right\|_F^2 = \sigma_x^2 d^2\left(1 + \rho(n-1)\right). \tag{17}$$

By making the additional assumption that the norm and the correlation propagate independently, the respective norm for the queries — and symmetrically the keys — (Eq. (7)) reduces to:

$$\mathbb{E}\left\|\frac{\partial\mathbf{S}}{\partial\mathbf{W}^Q}\right\|_F^2 = \sigma_x^6\frac{(n-1)}{n}(1-\rho)^2 d(n+d). \tag{18}$$

In Appendix A.2 we provide a rigorous proof, that relies on Isserlis theorem [Isserlis, 1918] to compute higher-order moments. The above expressions reveal the different dependencies on four main actors, that we inspect separately here. The gradients of the queries depend via a cubic function on the *variance of the input, $\sigma_x^2$*, compared to a linear for the values. This provides an additional interpretation of the successful use of layer normalization, as in Xiong et al. [2020], either in the POST-LN or PRE-LN format, that standardizes the input variance $\sigma_x^2$ to the value 1.

Next, we emphasize the dependence on the *correlation between the tokens*, also illustrated in Fig. 3. Importantly, note how the queries/keys have opposite monotonic functional dependence with respect to $\rho$ compared to the values. As revealed by Theorem 3.3 and Fig. 3 (center), inappropriate scaling of the residual branches can already lead to this phenomenon even in a relatively shallow network.

Finally, Eq. (17) and (18) reveal a different scaling in terms of the *embedding size $d$* and the *sequence length $n$* due to the self-attention operation itself. We hope that the identification of the different dependencies in the gradients of the parameters will inspire a new line of works aimed at solving some of the difficulties in training Transformers.

## 3.4    Connections to Optimization and Adaptive Methods

The existence of the discrepancy in the magnitude of the gradients with respect to the weights $\mathbf{W}^Q, \mathbf{W}^K$ and $\mathbf{W}^V$, might explain the success of adaptive optimization algorithms, as illustrated in Fig. 6, where we plot the effective learning rate computed by Adam [Kingma and Ba, 2014] in a toy encoder task (more details in Appendix C). Notice that the effective learning rate is increasingly larger (with depth) for the queries compared to the values — as postulated by our theory — and this difference is remarkably constant throughout training. Hence, we conjecture that the success of adaptive methods in Transformers' training can be partially explained by the need to fix this

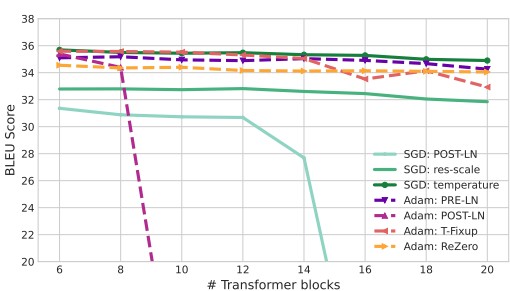

Figure 5: BLEU scores by increasing the number of transformers blocks. 'X' Transformer blocks implies in total 'X' encoder self-attention, 'X' decoder self-attention and 'X' decoder cross-attention layers.

Table 1: BLEU scores for the IWSLT14 German-to-English translation task. *SGD res-scale* refers to the training of SGD without layer normalization and initialization of the residual scaling $a_1 = a_2 = \frac{1}{\sqrt{L}}$. *SGD temperature* additionally employs an inverse temperature inside the softmax.

| Method (6L-Encoder / 6L-Decoder) | BLEU ↑ |
|---|---|
| SGD POST-LN | 31.36 |
| SGD res-scale | 32.79 |
| SGD temperature | **35.69** |
| Adam POST-LN [Vaswani et al., 2017] | 35.39 |
| Adam PRE-LN [Vaswani et al., 2017] | 35.10 |
| ReZero [Bachlechner et al., 2021] | 34.55 |
| T-Fixup Zhang et al. [2019] | 35.59 |

disproportionate gradient's magnitude. To test this hypothesis, we propose a simple architectural modification, an inverse temperature scaling $\tau \in \mathbb{R}$ inside the softmax:

$$\mathbf{S}_\tau^\ell := \text{softmax}\left(\frac{\tau}{\sqrt{d_k}}\mathbf{X}^\ell\mathbf{W}^Q\left(\mathbf{X}^\ell\mathbf{W}^K\right)^\top\right)\mathbf{X}^\ell\mathbf{W}^V. \tag{19}$$

A direct consequence of our analysis is that $\tau$ allows controlling the magnitude of the gradients for the queries and keys' parameters. In Section C.2, we detail how one can choose $\tau$ such that the magnitude of the gradients as derived in Equation 17 and 18 is approximately matched at initialization.

We evaluate our proposal, consisting of residual scaling and the aforementioned inverse temperature parameters, on the widely used IWSLT14 German-to-English (De-En) benchmark translation task. All details regarding the experimental setup and the choice of inverse temperature used are provided in Appendix C. We train a Transformer encoder-decoder of varying depth with stochastic gradient descent (SGD), after removing all normalization layers and adequately initializing the residual connections. For our training with SGD, we avoid using any learning rate warm-up, as commonly done for Adam, and instead use a step-scheduler to decrease the learning rate at 40% and 80% of training. We compare against the following methods that make use of Adam; POST-LN and PRE-LN refer to the aforementioned alternatives to apply layer normalization. We also compare against other successful techniques that rely on specific initializations to avoid layer normalization, such as ReZero [Bachlechner et al., 2021] and T-Fixup [Zhang et al., 2019]. We report the average BLEU score [Papineni et al., 2002] across 5 runs in Fig. 5 and Table 1.

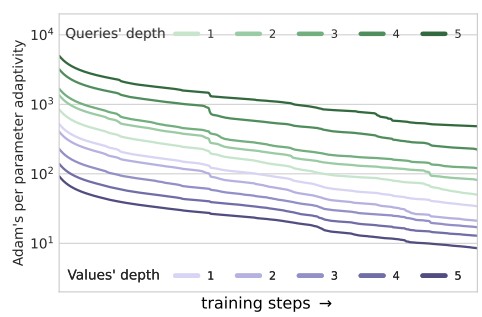

Figure 6: Adaptive learning rates computed by Adam in Transformers.

Our proposed method considerably improves training with SGD, keeping up and in some cases surpassing any results achieved by the Adam optimizer. We are also able to train deeper networks without the use of layer normalization. We leave for future work to further investigate modifications or alternatives to the self-attention operation.

## 4  Related Work

Our work builds upon the rich literature on forward and backward signal propagation in random neural networks [Poole et al., 2016, Schoenholz et al., 2017, Xiao et al., 2018, Pennington et al., 2017, Orvieto et al., 2021, Noci et al., 2021, Zavatone-Veth and Pehlevan, 2021]. The $1/\sqrt{L}$ scaling scheme has been investigated in the literature for the stabilization of residual networks [Hanin and Rolnick, 2018, Arpit et al., 2019, Allen-Zhu et al., 2019, Hayou et al., 2021].

Our work draws inspiration from a series of recent works studying the rank of the representations of random feed-forward neural networks at initialization [Daneshmand et al., 2020, 2021]. In the context of Transformers, Dong et al. [2021] has recently identified the rank collapse issue object of study of the present work. Thanks to our analysis of the backward pass, we are able to demonstrate that rank collapse in Transformer architectures leads to vanishingly small gradients of queries and keys, thereby preventing effective training and allowing us to complete the analysis of [Dong et al., 2021].

Among the architectural components in Transformers, layer normalization is, arguably, one of the most important – and debated – ones [Chen et al., 2018, Wang et al., 2019, Nguyen et al., 2010, Xiong et al., 2020]. In the original architecture [Vaswani et al., 2017], layer normalization is used to stabilize the forward pass by reducing the variance of the inputs to the following sublayer. Our analysis of the forward pass shows that its inclusion is not strictly necessary for the purpose of controlling the norm of the representations. For a theoretical analysis of signal propagation in the presence of layer norm, we refer the reader to Xiong et al. [2020].

Additionally, our theoretical study of the backward pass provides a rigorous explanation of the empirically observed discrepancy between the magnitude of the gradients of the queries and the values, which Liu et al. [2020] hypothesize to be one of the causes of the success of adaptive methods in training Transformers [Liu et al., 2019, Zhang et al., 2020, Huang et al., 2020].

Finally, properly rescaled residual connections have been found to be beneficial for training Transformers by a number of recent research works [Zhang et al., 2019, Bachlechner et al., 2021, Wang et al., 2022]. However, none of these studies characterize the impact of skip connections on rank propagation, while our analysis suggests a theoretically-grounded way to stabilize it.

## 5   Conclusions and Future Work

In this paper, we showed how, at initialization, rank collapse and more generally high correlation in the tokens causes vanishing gradients of the queries and keys of a Transformer architecture. While residual connections help mitigate rank collapse at finite depth, we showed that they alone cannot prevent high alignments of the tokens' representations — unless properly scaled by a $1/\sqrt{L}$-factor. Finally, we have also discovered counter-intuitive dependencies on the variance of the input, embedding size, and sequence length, potentially causing large differences between the gradients of queries/keys compared to the values' parameters. Hence, we conclude that one of the strengths of Transformers lies in their carefully designed architecture together with an adequate initialization. Finally, we gave preliminary evidence that one of the factors contributing to the higher efficacy of Adam compared to SGD in training Transformers arises from the disproportionate magnitude of gradients as postulated by our theory. Nonetheless, other factors might further accentuate the difference between these two algorithms during training, leaving the door open for further research regarding the benefits of adaptive optimization methods with Transformers.

## Acknowledgements

Sidak Pal Singh would like to acknowledge the financial support from Max Planck ETH Center for Learning Systems and the travel support from ELISE (GA no 951847).

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
