# Appendix

## Table of Contents

## A   Proof of Theorems

Recall the defining equations of a Transformer:

$$\mathbf{Z}^\ell = \alpha_1 \mathbf{S}^\ell + \mathbf{X}^\ell$$
$$\mathbf{Y}^\ell = \sigma(\mathbf{Z}^\ell \mathbf{W}^{F,1})\mathbf{W}^{F,2}$$
$$\mathbf{X}^{\ell+1} = \alpha_2 \mathbf{Y}^\ell + \mathbf{Z}^\ell,$$

where the self attention layers are defined as follows:

$$\mathbf{S}^\ell := \mathbf{A}^\ell \mathbf{X}^\ell \mathbf{W}^V, \text{ where } \mathbf{A}^\ell = \text{softmax}\left(\frac{1}{\sqrt{d_k}}\mathbf{X}^\ell \mathbf{W}^Q \left(\mathbf{X}^\ell \mathbf{W}^K\right)^\top\right).$$

We remark that when it is clear from the context, we suppress the $\ell$ index to improve readability.

**Initialization**: Recall that we initialize our weights with the so called "Xavier" [Glorot and Bengio, 2010] or "He" [He et al., 2015] initialization: each weight is sampled independently from a distribution with zero-mean and variance $\sigma_v^2 = \frac{1}{d_v}$ for the values and feedforward weights and $\sigma_k^2 = \frac{1}{d_k}$ for the queries and keys.

**Kronecker Delta**: we introduce the Kronecker Delta notation

$$\delta_{ab} = \begin{cases} 1 & a = b \\ 0 & a \neq b \end{cases}$$

and, similarly: $\delta_{a \neq b} = 1 - \delta_{ab}$.

**Notation**: in this section, we also adopt the following shorthand notation for the argument of the softmax $\mathbf{M} := \frac{\mathbf{X}\mathbf{W}^Q\mathbf{W}^{K\top}\mathbf{X}^\top}{\sqrt{d_k}}$. We will first compute the gradients with respect to values, queries and input (note that the gradients of the keys have the form as the queries, hence we omit the derivation). Recall that for a matrix $\mathbf{X}$, we use $\mathbf{X}_k$ to indicate its $k$-th row. Finally, we indicate with $\mathbf{1}_{n \times n}$ the $n \times n$ matrix with all ones, with $\mathbf{1}_n$ the columns vector with all ones, and with $\mathbf{I}_n$ the $n$-dimensional identity matrix.

## A.1 Backward Pass: Proofs of Lemma 3.1 and Theorem 3.2

In this section, we now look at the proofs of Lemma 3.1 and Theorem 3.2. We will first introduce our notation for the gradients, as well as some useful properties of the Kronecker product.

### A.1.1 Preliminaries

For the gradients, we avoid directly working with tensors by vectorizing the matrices in a *row-wise* fashion ($\mathrm{vec}_r$) and arranging the Jacobian in the numerator layout. More formally,

$$\frac{\partial \mathbf{Y}}{\partial \mathbf{X}} := \frac{\partial \, \mathrm{vec}_r(\mathbf{Y})}{\partial \, \mathrm{vec}_r(\mathbf{X})^\top} \, .$$

Alongside this, we use the following rule ($\otimes$ is the Kronecker product):

$$\frac{\partial \mathbf{AWB}}{\partial \mathbf{W}} = \mathbf{A} \otimes \mathbf{B}^\top \, . \tag{20}$$

For the proof of this rule, we refer to Singh et al. [2021], and to Magnus and Neudecker [2019] for a complete introduction to matrix calculus. We will also use the following well-known properties of the Kronecker product.

**Lemma A.1.** *Given the matrices* $\mathbf{A} \in \mathbb{R}^{n \times m}$, $\mathbf{B} \in \mathbb{R}^{p \times q}$, $\mathbf{C} \in \mathbb{R}^{m \times r}$, $\mathbf{D} \in \mathbb{R}^{q \times s}$, *then the following holds:*

$$\mathrm{tr}(\mathbf{A} \otimes \mathbf{B}) = \mathrm{tr}(\mathbf{A}) \, \mathrm{tr}(\mathbf{B}), \tag{21}$$

*and*

$$(\mathbf{A} \otimes \mathbf{B})(\mathbf{C} \otimes \mathbf{D}) = (\mathbf{AC}) \otimes (\mathbf{BD}). \tag{22}$$

### A.1.2 Proof of Lemma 3.1

In Lemma A.2 and Lemma A.3 we compute the gradients with respect to the queries, values and $\mathbf{X}$, respectively. Then we use these results to prove Lemma 3.1 by computing the expectation of the Frobenius norms.

**Lemma A.2** (Gradients of Self Attention for parameter matrices). *The gradients of the self attention layer defined in Eq. (1) have the following form:*

$$\frac{\partial \mathbf{S}}{\partial \mathbf{W}^V} = \mathrm{softmax}\left(\frac{\mathbf{XW}^Q \mathbf{W}^{K^\top} \mathbf{X}^\top}{\sqrt{d_k}}\right) \mathbf{X} \otimes \mathbf{I}_{d_v}$$

$$\frac{\partial \mathbf{S}}{\partial \mathbf{W}^Q} = \left(\mathbf{I}_n \otimes \mathbf{W}^{V^\top} \mathbf{X}^\top\right) \frac{\partial \mathbf{A}}{\partial \mathbf{M}} \left(\frac{\mathbf{X} \otimes \mathbf{XW}^K}{\sqrt{d_k}}\right),$$

*where the gradients of the softmax with respect to its inputs are as follows:*

$$\frac{\partial \mathbf{A}}{\partial \mathbf{M}} = \mathrm{blockdiag}\left(\frac{\partial \mathbf{A}_i}{\partial \mathbf{M}_i^\top}\right) \tag{23}$$

*and where* $\frac{\partial \mathbf{A}_i}{\partial \mathbf{M}_i^\top} = \mathrm{diag}(\mathbf{A}_i) - \mathbf{A}_i \mathbf{A}_i^\top$ *with* $\mathbf{A}_i$ *being the* $i$-th *row of* $\mathbf{A}$ *in column vector format.*
*Finally, note that under the uniform-attention assumption, Eq. (23) simplifies to:*

$$\frac{\partial \mathbf{A}}{\partial \mathbf{M}} = \frac{1}{n}\mathbf{I}_n \otimes \left(\mathbf{I}_n - \frac{1}{n}\mathbf{1}_{n \times n}\right). \tag{24}$$

*Proof.* Let's start with the simple case of the values' weights $\mathbf{W}^V$. Using the rule in Eq. (20), it is immediate that:

$$\frac{\partial \mathbf{S}}{\partial \mathbf{W}^V} = \mathrm{softmax}\left(\frac{\mathbf{XW}^Q \mathbf{W}^{K^\top} \mathbf{X}^\top}{\sqrt{d_k}}\right) \mathbf{X} \otimes \mathbf{I}_{d_v} = \mathbf{AX} \otimes \mathbf{I}_{d_v} \, .$$

For the queries, a simple application of the chain rule and then again Eq. (20) gives:

$$\frac{\partial \mathbf{S}}{\partial \mathbf{W}^Q} = \frac{\partial \mathbf{S}}{\partial \mathbf{A}} \frac{\partial \mathbf{A}}{\partial \mathbf{W}^Q} = \frac{\partial \mathbf{S}}{\partial \mathbf{A}} \frac{\partial \mathbf{A}}{\partial \mathbf{M}} \frac{\partial \mathbf{M}}{\partial \mathbf{W}^Q}$$

$$= \left( \mathbf{I}_n \otimes \mathbf{W}^{V\top} \mathbf{X}^\top \right) \frac{\partial \mathbf{A}}{\partial \mathbf{M}} \left( \frac{\mathbf{X} \otimes \mathbf{X} \mathbf{W}^K}{\sqrt{d_k}} \right),$$

which is the desired results. Finally, for the gradients of the softmax note that:

$$\frac{\partial \mathbf{A}_{pq}}{\partial \mathbf{M}_{ij}} = \frac{\partial}{\partial \mathbf{M}_{ij}} \frac{\exp(\mathbf{M}_{pq})}{\sum_k \exp(\mathbf{M}_{pk})} = \delta_{ip}\delta_{jq}\mathbf{A}_{ij} - \delta_{ip}\mathbf{A}_{iq}\mathbf{A}_{ij}.$$

By writing the above expression in the matrix notation described above, we obtain the desired result. More specifically, the block diagonal structure is given from the term $\delta_{ip}$ which stems from the fact that the softmax is applied row-wise. $\square$

---

**Lemma A.3** (Gradients of Self Attention with respect to the Embedding matrix). *The gradients of the self attention layer with respect to the embedding matrix $\mathbf{X}$ defined in Eq. (1) have the following form*

$$\frac{\partial \mathbf{S}}{\partial \mathbf{X}} = \frac{1}{\sqrt{d_k}}(\mathbf{I}_n \otimes \mathbf{W}^{\mathbf{V}\top}\mathbf{X}^\top)\frac{\partial \mathbf{A}}{\partial \mathbf{M}} \left( \mathbf{I}_n \otimes \mathbf{X}\mathbf{W}^K\mathbf{W}^{\mathbf{Q}\top} + \mathbf{K}_{nn}(\mathbf{I}_n \otimes \mathbf{X}\mathbf{W}^Q\mathbf{W}^{\mathbf{K}\top}) \right) + \mathbf{A} \otimes \mathbf{W}^{V\top}, \tag{25}$$

*where the gradients of the softmax with respect to its inputs are denoted by $\frac{\partial \mathbf{A}}{\partial \mathbf{M}}$ as before.*

---

*Proof.* Remember that we defined $\mathbf{S} = \mathrm{softmax}(\frac{1}{\sqrt{d_k}}\mathbf{X}\mathbf{W}^Q\mathbf{W}^{\mathbf{K}\top}\mathbf{X}^\top)\mathbf{X}\mathbf{W}^V$. Alongside with our previous shorthands $\mathbf{A}$, $\mathbf{M}$, let us define the remaining $\mathbf{X}\mathbf{W}^V$ as a matrix $\mathbf{T}$, so that $\mathbf{S} = \mathbf{A}\,\mathbf{T}$. Both $\mathbf{A}$ and $\mathbf{T}$ are functions of $\mathbf{X}$. So the matrix differential can be written as:

$$\frac{\partial \mathbf{S}}{\partial \mathbf{X}} = \frac{\partial \mathbf{S}}{\partial \mathbf{A}} \frac{\partial \mathbf{A}}{\partial \mathbf{X}} + \frac{\partial \mathbf{S}}{\partial \mathbf{T}} \frac{\partial \mathbf{T}}{\partial \mathbf{X}} \tag{26}$$

$$= \frac{\partial \mathbf{S}}{\partial \mathbf{A}} \frac{\partial \mathbf{A}}{\partial \mathbf{M}} \frac{\partial \mathbf{M}}{\partial \mathbf{X}} + \frac{\partial \mathbf{S}}{\partial \mathbf{T}} \frac{\partial \mathbf{T}}{\partial \mathbf{X}} \tag{27}$$

$$= (\mathbf{I}_n \otimes \mathbf{W}^{\mathbf{V}\top}\mathbf{X}^\top) \frac{\partial \mathbf{A}}{\partial \mathbf{M}} \frac{\partial \mathbf{M}}{\partial \mathbf{X}} + (\mathbf{A} \otimes \mathbf{I}_d)(\mathbf{I}_n \otimes \mathbf{W}^{\mathbf{V}\top}) \tag{28}$$

$$= (\mathbf{I}_n \otimes \mathbf{W}^{\mathbf{V}\top}\mathbf{X}^\top) \frac{\partial \mathbf{A}}{\partial \mathbf{M}} \frac{\partial \mathbf{M}}{\partial \mathbf{X}} + (\mathbf{A} \otimes \mathbf{W}^{\mathbf{V}\top}) \tag{29}$$

Next, we use the matrix differential and then the identification theorem of matrix derivatives to compute the matrix gradient $\frac{\partial \mathbf{A}}{\partial \mathbf{X}}$

$$d\mathbf{A} = \frac{1}{\sqrt{d_k}}d(\mathbf{X})\,\mathbf{W}^Q\mathbf{W}^{\mathbf{K}\top}\mathbf{X}^\top + \frac{1}{\sqrt{d_k}}\mathbf{X}\mathbf{W}^Q\mathbf{W}^{\mathbf{K}\top}\,d(\mathbf{X}^\top).$$

Vectorizing both sides:

$$d\,\mathrm{vec}_r(\mathbf{A}) = \frac{1}{\sqrt{d_k}}(\mathbf{I}_n \otimes \mathbf{X}\mathbf{W}^K\mathbf{W}^{\mathbf{Q}\top})d(\mathrm{vec}_r(\mathbf{X})) + \frac{1}{\sqrt{d_k}}(\mathbf{X}\mathbf{W}^Q\mathbf{W}^{\mathbf{K}\top} \otimes \mathbf{I}_n)\,d(\mathrm{vec}_r(\mathbf{X}^\top))$$

$$= \frac{1}{\sqrt{d_k}}(\mathbf{I}_n \otimes \mathbf{X}\mathbf{W}^K\mathbf{W}^{\mathbf{Q}\top})d(\mathrm{vec}_r(\mathbf{X})) + \frac{1}{\sqrt{d_k}}(\mathbf{X}\mathbf{W}^Q\mathbf{W}^{\mathbf{K}\top} \otimes \mathbf{I}_n)\mathbf{K}_{dn}\,d(\mathrm{vec}_r(\mathbf{X})).$$

Recall, for an arbitrary matrix $\mathbf{B} \in \mathbb{R}^{m \times n}$, the commutation matrix $\mathbf{K}_{mn}$ transforms columnwise vectorization into rowwise vectorization. More precisely,

$$\mathbf{K}_{mn}\,\mathrm{vec}_c(\mathbf{B}) = \mathrm{vec}_c(\mathbf{B}^\top)$$

and $\mathrm{vec}_c(\mathbf{B}) = \mathrm{vec}_r(\mathbf{B}^\top)$. Therefore, for rowwise vectorization, we have a similar result:

$$\mathbf{K}_{mn}\,\mathrm{vec}_r(\mathbf{B}^\top) = \mathrm{vec}_r(\mathbf{B})$$

$$\mathrm{vec}_r(\mathbf{B}^\top) = \mathbf{K}_{nm}\,\mathrm{vec}_r(\mathbf{B}),$$

where in the last line we used the fact the commutation is a permutation matrix, so $\mathbf{K}_{mn}^{-1} = \mathbf{K}_{mn}^{\top} = \mathbf{K}_{nm}$. Thus, we get the required matrix derivative as follows:

$$\frac{\partial \mathbf{A}}{\partial \mathbf{X}} = \frac{1}{\sqrt{d_k}} \mathbf{I}_n \otimes \mathbf{X}\mathbf{W}^K \mathbf{W}^{\mathbf{Q}\top} + \frac{1}{\sqrt{d_k}} (\mathbf{X}\mathbf{W}^Q \mathbf{W}^{\mathbf{K}\top} \otimes \mathbf{I}_n) \mathbf{K}_{dn} .$$

Next, we will use a property of commutation matrix to make things simpler (Theorem 7.9, Magnus and Neudecker [2019]):

$$\frac{\partial \mathbf{A}}{\partial \mathbf{X}} = \frac{1}{\sqrt{d_k}} \mathbf{I}_n \otimes \mathbf{X}\mathbf{W}^K \mathbf{W}^{\mathbf{Q}\top} + \frac{1}{\sqrt{d_k}} \mathbf{K}_{nn}(\mathbf{I}_n \otimes \mathbf{X}\mathbf{W}^Q \mathbf{W}^{\mathbf{K}\top}).$$

Plugging this into the above Eq. (29), we get:

$$\frac{\partial \mathbf{S}}{\partial \mathbf{X}} = \frac{1}{\sqrt{d_k}} (\mathbf{I}_n \otimes \mathbf{W}^{\mathbf{V}\top}\mathbf{X}^{\top}) \frac{\partial \mathbf{A}}{\partial \mathbf{M}} \left( \mathbf{I}_n \otimes \mathbf{X}\mathbf{W}^K \mathbf{W}^{\mathbf{Q}\top} + \mathbf{K}_{nn}(\mathbf{I}_n \otimes \mathbf{X}\mathbf{W}^Q \mathbf{W}^{\mathbf{K}\top}) \right) + \mathbf{A} \otimes \mathbf{W}^{V\top}.$$

As a sanity check, we can calculate if the shapes of the matrices are consistent. LHS should be a $nd \times nd$ matrix, while the constituent matrices of the first term on RHS: $\mathbf{I}_n \otimes \mathbf{W}^{V\top}\mathbf{X}^{\top} \in \mathbb{R}^{nd \times n^2}$, $\frac{\partial \mathbf{A}}{\partial \mathbf{M}} \in \mathbb{R}^{n^2 \times n^2}$, the additive term next to it is a $n^2 \times nd$ matrix, and the second term on RHS is a Kronecker product of a $n \times n$ and a $d \times d$ matrix. □

---

**Lemma 3.1.** *Let $\mathbf{X}^{\ell}$ be the representations of the input sequence at the $\ell$-th layer. Under the uniform-attention assumption, we have*

$$\mathbb{E}\left\| \frac{\partial \mathbf{S}^{\ell}}{\partial \mathbf{W}^{V,\ell}} \right\|_F^2 = d_v n \mathbb{E}\|\bar{\boldsymbol{x}}^{\ell}\|^2 ; \tag{6}$$

$$\mathbb{E}\left\| \frac{\partial \mathbf{S}^{\ell}}{\partial \mathbf{W}^{Q,\ell}} \right\|_F^2 = \frac{\sigma_v^2 \sigma_k^2 d_v}{n^2} \cdot \mathbb{E}\left[ \|\mathbf{X}^{\ell}\|_F^2 \cdot \|(\mathbf{X}^{\ell})^{\top}\mathbf{X}^{\ell} - n\bar{\boldsymbol{x}}^{\ell}(\bar{\boldsymbol{x}}^{\ell})^{\top}\|_F^2 \right] ; \tag{7}$$

$$\mathbb{E}\left\| \frac{\partial \mathbf{S}^{\ell}}{\partial \mathbf{X}^{\ell}} \right\|_F^2 \leq \frac{8\sigma_q^2 \sigma_k^2 \sigma_v^2 d_k d_v}{n} \cdot \mathbb{E}\left\| (\mathbf{X}^{\ell})^{\top}\mathbf{X}^{\ell} - n\bar{\boldsymbol{x}}^{\ell}(\bar{\boldsymbol{x}}^{\ell})^{\top} \right\|_F^2 + 2d_v^2 \sigma_v^2 . \tag{8}$$

---

*Proof.* Here, we suppress the index $\ell$.

**Gradient with respect to the values matrix.**

Recall that from Lemma A.2 we have that:

$$\frac{\partial \mathbf{S}}{\partial \mathbf{W}^V} = \mathrm{softmax}\left( \frac{\mathbf{X}\mathbf{W}^Q \mathbf{W}^{K\top}\mathbf{X}^{\top}}{\sqrt{d_k}} \right) \mathbf{X} \otimes \mathbf{I}_{d_v} \overset{\text{Ass. 3.1}}{=} \frac{1}{n} \mathbf{1}_{n \times n} \mathbf{X} \otimes \mathbf{I}_{d_v}.$$

By direct computation:

$$\left\| \frac{\partial \mathbf{S}}{\partial \mathbf{W}^V} \right\|_F^2 = \mathrm{tr}\left( \frac{\partial \mathbf{S}}{\partial \mathbf{W}^V} \frac{\partial \mathbf{S}}{\partial \mathbf{W}^V}^{\top} \right) \overset{(22)}{=} \frac{1}{n^2} \mathrm{tr}((\mathbf{1}_{n \times n}\mathbf{X}\mathbf{X}^{\top}\mathbf{1}_{n \times n}) \otimes \mathbf{I}_{d_v})$$

$$\overset{(21)}{=} \frac{1}{n^2} \mathrm{tr}(\mathbf{1}_{n \times n}\mathbf{X}\mathbf{X}^{\top}\mathbf{1}_{n \times n}) \mathrm{tr}(\mathbf{I}_{d_v})$$

$$= \frac{d}{n^2} \mathrm{tr}(\mathbf{X}\mathbf{X}^{\top}\mathbf{1}_{n \times n}\mathbf{1}_{n \times n})$$

$$= \frac{d_v}{n} \mathrm{tr}(\mathbf{X}\mathbf{X}^{\top}\mathbf{1}_{n \times n})$$

$$= \frac{d_v}{n} \mathrm{tr}(\mathbf{X}\mathbf{X}^{\top}\mathbf{1}_n\mathbf{1}_n^{\top})$$

$$= \frac{d_v}{n} \mathbf{1}_n^{\top}\mathbf{X}\mathbf{X}^{\top}\mathbf{1}_n$$

$$= \frac{d_v}{n} \|\mathbf{X}^{\top}\mathbf{1}_n\|^2 = d_v n \|\bar{\boldsymbol{x}}\|^2 .$$

**Gradients with respect to the queries/keys matrix.**

First, recall the expression for the gradient of the softmax under the uniform-attention assumption (Eq. (24)):

$$\frac{\partial \mathbf{A}}{\partial \mathbf{M}} = \frac{1}{n}\mathbf{I}_n \otimes \left(\mathbf{I}_n - \frac{1}{n}\mathbf{1}_{n\times n}\right).$$

Hence, we can rewrite the expression of Lemma A.2 for the gradients of the queries as:

$$\frac{\partial \mathbf{S}}{\partial \mathbf{W}^Q} = \left(\mathbf{I}_n \otimes \mathbf{W}^{V\top}\mathbf{X}^\top\right)\frac{\partial \mathbf{A}}{\partial \mathbf{M}}\left(\frac{\mathbf{X}\otimes\mathbf{X}\mathbf{W}^K}{\sqrt{d_k}}\right)$$

$$= \frac{1}{\sqrt{d_k}n}\left(\mathbf{I}_n \otimes \mathbf{W}^{V\top}\mathbf{X}^\top\right)\left[\mathbf{I}_n \otimes \left(\mathbf{I}_n - \frac{1}{n}\mathbf{1}_{n\times n}\right)\right]\left(\mathbf{X}\otimes\mathbf{X}\mathbf{W}^K\right)$$

$$= \frac{1}{\sqrt{d_k}n}\mathbf{X}\otimes\left[\mathbf{W}^{V\top}\mathbf{X}^\top\left(\mathbf{I}_n - \frac{1}{n}\mathbf{1}_{n\times n}\right)\mathbf{X}\mathbf{W}^K\right],$$

where in the last step we have used twice the property of the Kronecker product in Eq. (22) of Lemma A.1.

Hence,

$$\left\|\frac{\partial \mathbf{S}}{\partial \mathbf{W}^Q}\right\|_F^2 = \mathrm{tr}\left(\frac{\partial \mathbf{S}}{\partial \mathbf{W}^Q}\frac{\partial \mathbf{S}}{\partial \mathbf{W}^Q}^T\right) \stackrel{(21)}{=} \frac{1}{d_k n^2}\|\mathbf{X}\|_F^2 \cdot \left\|\mathbf{W}^{V\top}\mathbf{X}^\top\left(\mathbf{I}_n - \frac{1}{n}\mathbf{1}_{n\times n}\right)\mathbf{X}\mathbf{W}^K\right\|_F^2,$$

where we have used the property on the trace of the Kronecker product (Lemma A.1, Eq. (21)). Note that if we are conditioning on $\mathbf{X}$, then we only have to take the expectation of the last term with respect to the weights $\mathbf{W}^K$ and $\mathbf{W}^V$. Let us call $\mathbf{L} := \mathbf{I}_n - \frac{1}{n}\mathbf{1}_{n\times n}$ for notation simplicity.

Note: for a matrix $\mathbf{W} \in \mathbb{R}^{d\times d}$ whose entries $w_{ij} \sim \mathcal{N}(0, \sigma^2)$, then $\mathbb{E}\mathbf{W}\mathbf{W}^\top = d\sigma^2\,\mathbf{I}_d$. Thus, exchanging the order of trace and expectation, we can write:

$$\mathbb{E}\|\mathbf{W}^{\mathbf{V}\top}\mathbf{X}^\top\mathbf{L}\mathbf{X}\mathbf{W}^K\|_F^2 = \mathbb{E}\,\mathrm{tr}(\mathbf{W}^{\mathbf{V}\top}\mathbf{X}^\top\mathbf{L}\mathbf{X}\mathbf{W}^K \cdot \mathbf{W}^{\mathbf{K}\top}\mathbf{X}^\top\mathbf{L}\mathbf{X}\mathbf{W}^V)$$

$$= \mathrm{tr}(\mathbf{X}^\top\mathbf{L}\mathbf{X}\mathbb{E}[\mathbf{W}^K\mathbf{W}^{\mathbf{K}\top}]\mathbf{X}^\top\mathbf{L}\mathbf{X}\mathbb{E}[\mathbf{W}^V\mathbf{W}^{\mathbf{V}\top}])$$

$$= \sigma_v^2\sigma_k^2 d_k d_v\,\mathrm{tr}(\mathbf{X}^\top\mathbf{L}\mathbf{X}\cdot\mathbf{X}^\top\mathbf{L}\mathbf{X})$$

$$= \sigma_v^2\sigma_k^2 d_k d_v\|\mathbf{X}^\top\mathbf{L}\mathbf{X}\|_F^2$$

$$= \sigma_v^2\sigma_k^2 d_k d_v\left\|\mathbf{X}^\top(\mathbf{I}_n - \frac{1}{n}\mathbf{1}_n\mathbf{1}_n^\top)\mathbf{X}\right\|_F^2$$

$$= \sigma_v^2\sigma_k^2 d_k d_v\|\mathbf{X}^\top\mathbf{X} - n\bar{\boldsymbol{x}}\bar{\boldsymbol{x}}^\top\|_F^2,$$

where, $\bar{\boldsymbol{x}} = \frac{1}{n}\mathbf{X}^\top\mathbf{1}_n \in \mathbb{R}^d$ is the mean embedding. Multiply this by $\frac{1}{d_k n^2}\|\mathbf{X}\|_F^2$ to get the final answer.

**Gradient with respect to the input.**

Plugging in the values of $\frac{\partial \mathbf{A}}{\partial \mathbf{M}}$ and $\mathbf{A}$ under the uniform-attention assumption into Eq. (25) gives rise to the following:

$$\frac{\partial \mathbf{S}}{\partial \mathbf{X}} = \frac{1}{n\sqrt{d_k}}\mathbf{I}_n \otimes \mathbf{W}^{\mathbf{V}\top}\mathbf{X}^\top\left(\mathbf{I}_n - \frac{1}{n}\mathbf{1}_n\mathbf{1}_n^\top\right)\mathbf{X}\mathbf{W}^K\mathbf{W}^{\mathbf{Q}\top}$$

$$+ \frac{1}{n\sqrt{d_k}}\left[\mathbf{I}_n \otimes \mathbf{W}^{\mathbf{V}\top}\mathbf{X}^\top\left(\mathbf{I}_n - \frac{1}{n}\mathbf{1}_n\mathbf{1}_n^\top\right)\right]\mathbf{K}_{nn}(\mathbf{I}_n \otimes \mathbf{X}\mathbf{W}^Q\mathbf{W}^{\mathbf{K}\top})$$

$$+ \frac{1}{n}\mathbf{1}_{n\times n}\otimes\mathbf{W}^{\mathbf{V}\top}$$

Let's refer to the matrices on the right-hand side as $\mathbf{A}_1, \mathbf{A}_2, \mathbf{A}_3$ respectively. We compute the expected squared Frobenius norm of these as follows:

For $\mathbf{A}_3$:

$$\mathbb{E}[\|\mathbf{A}_3\|_F^2] = \frac{1}{n^2}\mathbb{E}[\mathrm{tr}(n\mathbf{1}_{n\times n}\otimes\mathbf{W}^{\mathbf{V}\top}\mathbf{W}^V)]$$

$$\stackrel{(21)}{=} \frac{1}{n}\mathrm{tr}(\mathbf{1}_{n\times n})\,\mathrm{tr}(\mathbb{E}[\mathbf{W}^V\mathbf{W}^{\mathbf{V}\top}]) = d_v^2\sigma_v^2.$$

Similarly, for $\mathbf{A}_1$:

$$\mathbb{E}[\|\mathbf{A}_1\|_F^2] = \frac{\sigma_q^2 \sigma_k^2 \sigma_v^2 d_k d_v}{n^2} \operatorname{tr}(\mathbf{I}_n) \operatorname{tr}(\mathbf{X}^\top \mathbf{L} \mathbf{X} \cdot \mathbf{X}^\top \mathbf{L} \mathbf{X}) \tag{30}$$

$$= \frac{\sigma_q^2 \sigma_k^2 \sigma_v^2 d_k d_v}{n} \left\| \mathbf{X}^\top \mathbf{X} - n \bar{\boldsymbol{x}} \bar{\boldsymbol{x}}^\top \right\|_F^2 \tag{31}$$

$$= \sigma_q^2 \sigma_k^2 \sigma_v^2 d_k d_v n \left\| \frac{1}{n} \mathbf{X}^\top \mathbf{X} - \bar{\boldsymbol{x}} \bar{\boldsymbol{x}}^\top \right\|_F^2 . \tag{32}$$

Finally, for $\mathbf{A}_2$:

$$\mathbb{E}[\|\mathbf{A}_2\|_F^2] \tag{33}$$

$$= \frac{1}{n^2 d_k} \mathbb{E}\left[ \operatorname{tr}\!\left( \left[ \mathbf{I}_n \otimes \mathbf{W}^{\mathbf{V}^\top} \mathbf{X}^\top \mathbf{L} \right] \mathbf{K}_{nn} (\mathbf{I}_n \otimes \mathbf{X} \mathbf{W}^Q \mathbf{W}^{\mathbf{K}^\top} \mathbf{W}^K \mathbf{W}^{\mathbf{Q}^\top} \mathbf{X}^\top) \mathbf{K}_{nn} \left[ \mathbf{I}_n \otimes \mathbf{L} \mathbf{X} \mathbf{W}^V \right] \right) \right] \tag{34}$$

$$= \frac{\sigma_q^2 \sigma_k^2 \sigma_v^2 d_k d_v}{n^2} \operatorname{tr}((\mathbf{I}_n \otimes \mathbf{X} \mathbf{X}^\top)[\mathbf{I}_n \otimes \mathbf{L} \mathbf{X} \mathbf{X}^\top \mathbf{L}]) \tag{35}$$

$$= \frac{\sigma_q^2 \sigma_k^2 \sigma_v^2 d_k d_v}{n} \operatorname{tr}(\mathbf{X}^\top \mathbf{L} \mathbf{X} \cdot \mathbf{X}^\top \mathbf{L} \mathbf{X}) = \mathbb{E}[\|\mathbf{A}_1\|_F^2], \tag{36}$$

where in the second line we have taken the expectation inside and used the fact that $\mathbf{K}_{nn}$, being a commutation matrix, is orthogonal. Then, by simple properties of Kronecker product and cyclic property of trace, we have the result, which is the same as that for $\mathbf{A}_1$.

Finally, by the triangle inequality

$$\mathbb{E}\left\| \frac{\partial \mathbf{S}}{\partial \mathbf{X}} \right\|^2 \leq 2\mathbb{E}\|\mathbf{A}_1 + \mathbf{A}_2\|^2 + 2\mathbb{E}\|\mathbf{A}_3\|^2 \tag{37}$$

$$\leq 4\mathbb{E}\|\mathbf{A}_1\|^2 + 4\mathbb{E}\|\mathbf{A}_2\|^2 + 2\mathbb{E}\|\mathbf{A}_3\|^2 \tag{38}$$

$$= 8\mathbb{E}\|\mathbf{A}_1\|^2 + 2\mathbb{E}\|\mathbf{A}_3\|^2 \tag{39}$$

$$= \frac{8\sigma_q^2 \sigma_k^2 \sigma_v^2 d_k d_v}{n} \left\| \mathbf{X}^\top \mathbf{X} - n \bar{\boldsymbol{x}} \bar{\boldsymbol{x}}^\top \right\|_F^2 + 2 d_v^2 \sigma_v^2. \tag{40}$$

This completes the proof. $\qquad\square$

### A.1.3  Proof of Theorem 3.2

**Theorem 3.2** (Vanishing gradients under rank collapse).  *Suppose that the uniform-attention assumption holds. If additionally $\mathbf{X}^\ell$ for any $l \in [L]$ has rank-1, and there exists a vector $\boldsymbol{x} \in \mathbb{R}^d$ such that $\mathbf{X}^\ell = \mathbf{1}_n \boldsymbol{x}^T$, then:*

$$\mathbb{E}\left\| \frac{\partial \mathcal{L}}{\partial \mathbf{W}^{Q,\ell}} \right\|_F^2 = 0, \quad \mathbb{E}\left\| \frac{\partial \mathcal{L}}{\partial \mathbf{W}^{K,\ell}} \right\|_F^2 = 0, \tag{9}$$

*where the expectation is taken over the weight matrices. This implies that these quantities are vanishing almost surely, due to the non-negativeness of the norm.*

Before starting the proof, it is interesting to note that, even though the gradients of queries and keys vanish in the rank collapse regime (i.e. $\left\| \mathbf{X}^\top \mathbf{X} - n \bar{\boldsymbol{x}} \bar{\boldsymbol{x}}^\top \right\| = 0$), the gradient with respect to the values and the input does not (see Theorem 3.1). From this simple remark, we can conclude that, even in the rank collapse regime, information still propagates in the backward pass. In Section 3.4 (main paper), we show that even if gradients effectively propagate, the phenomenon studied in this theorem still greatly affects training.

*Proof.* By using the chain rule and the fact that for two matrixes $\mathbf{A}, \mathbf{B}$ we have that $\|\mathbf{A}\mathbf{B}\|_F^2 \leq \|\mathbf{A}\|_F^2 \|\mathbf{B}\|_F^2$, we can upper bound the gradient as:

$$
\left\| \frac{\partial \mathcal{L}}{\partial \mathbf{W}^{Q,\ell}} \right\|_F^2 \leq \prod_{i=\ell+1}^{L-1} \left\| \frac{\partial \mathbf{X}^{i+1}}{\partial \mathbf{X}^i} \right\|_F^2 \left\| \frac{\partial \mathcal{L}}{\partial \mathbf{X}^L} \right\|_F^2 \left\| \frac{\partial \mathbf{X}^{\ell+1}}{\partial \mathbf{W}^{Q,\ell}} \right\|_F^2
$$

$$
\leq \prod_{i=\ell+1}^{L-1} \left\| \frac{\partial \mathbf{X}^{i+1}}{\partial \mathbf{X}^i} \right\|_F^2 \left\| \frac{\partial \mathcal{L}}{\partial \mathbf{X}^L} \right\|_F^2 \left\| \frac{\partial \mathbf{X}^{\ell+1}}{\partial \mathbf{Z}^\ell} \right\|_F^2 \left\| \frac{\partial \mathbf{Z}^\ell}{\partial \mathbf{W}^{Q,\ell}} \right\|_F^2
$$

$$
\leq \prod_{i=\ell+1}^{L-1} \left\| \frac{\partial \mathbf{X}^{i+1}}{\partial \mathbf{X}^i} \right\|_F^2 \left\| \frac{\partial \mathcal{L}}{\partial \mathbf{X}^L} \right\|_F^2 \left\| \frac{\partial \mathbf{X}^{\ell+1}}{\partial \mathbf{Z}^\ell} \right\|_F^2 \left( \left\| \frac{\partial \alpha_1 \mathbf{S}^\ell}{\partial \mathbf{W}^{Q,\ell}} \right\|_F^2 + \underbrace{\left\| \frac{\partial \mathbf{X}^\ell}{\partial \mathbf{W}^{Q,\ell}} \right\|_F^2}_{=0} \right),
$$

where we recall that $\mathbf{Z}^\ell = \alpha_1 \mathbf{S}^\ell + \mathbf{X}^\ell$ and in the last step we have used that $\mathbf{X}^\ell$ does not depend on $\mathbf{W}^{Q,\ell}$, hence the gradient vanishes. By taking expectation and using the tower property, we have that:

$$
\mathbb{E} \left\| \frac{\partial \mathcal{L}}{\partial \mathbf{W}^{Q,\ell}} \right\|_F^2 \leq \mathbb{E} \left[ \underbrace{\mathbb{E} \left[ \prod_{i=\ell+1}^{L-1} \left\| \frac{\partial \mathbf{X}^{i+1}}{\partial \mathbf{X}^i} \right\|_F^2 \left\| \frac{\partial \mathcal{L}}{\partial \mathbf{X}^L} \right\|_F^2 \left\| \frac{\partial \mathbf{X}^{\ell+1}}{\partial \mathbf{Z}^\ell} \right\|_F^2 \right]}_{=:G(\mathbf{X}^\ell)} \left\| \frac{\partial \alpha_1 \mathbf{S}^\ell}{\partial \mathbf{W}^{Q,\ell}} \right\|_F^2 \right],
$$

where the expectations are taken with respect to $\mathbf{X}^\ell$ for the outer one and conditioning on $\mathbf{X}^\ell$ for inner one. Indeed, the first three terms only depend on the network values after $\mathbf{X}^\ell$. Now, a repeated application of the tower property in $G(\mathbf{X}^\ell)$, together with the results on the gradients of Lemma 3.1, easily shows that $G(\mathbf{X}^\ell)$ stays bounded under our hypothesis. To see this one can also simply note that, since the softmax and its derivatives are almost surely bounded, the boundedness of $G(\mathbf{X}^\ell)$ is implied by an analogous statement for a vanilla linear MLP (i.e removing the softmax). In this setting, the random variable inside the expectation in $G(\mathbf{X}^\ell)$ is a finite linear combination of Gaussian products — which has bounded expectation.

All in all, we have that

$$
\mathbb{E} \left\| \frac{\partial \mathcal{L}}{\partial \mathbf{W}^{Q,\ell}} \right\|_F^2 \leq \mathbb{E} \left[ B_{\mathbf{X}^\ell} \left\| \frac{\partial \alpha_1 \mathbf{S}^\ell}{\partial \mathbf{W}^{Q,\ell}} \right\|_F^2 \right],
$$

where $B_{\mathbf{X}^\ell}$ is an almost-surely-bounded function of $\mathbf{X}^\ell$. Hence, to show that $\mathbb{E} \left\| \frac{\partial \mathcal{L}}{\partial \mathbf{W}^{Q,\ell}} \right\|_F^2 = 0$, we now just need to show that:

$$
\mathbb{E} \left\| \frac{\partial \alpha_1 \mathbf{S}^\ell}{\partial \mathbf{W}^{Q,\ell}} \right\|_F^2 = 0
$$

under the rank-1 hypothesis for $\mathbf{X}^\ell$. Let $\mathbf{X}_1^\ell, \dots \mathbf{X}_n^\ell \in \mathbb{R}^{d_v}$ be the representations for the $n$ tokens. Under the rank-1 assumption, each token can be written as a multiple of a single vector $\boldsymbol{x} \in \mathbb{R}^{d_v}$, and hence there exists $a_1, \dots, a_n \in \mathbb{R}$ such that $\mathbf{X}_1 = a_1 \boldsymbol{x}, \dots, \mathbf{X}_n = a_n \boldsymbol{x}$. From Lemma 3.1, we know that:

$$
\mathbb{E} \left\| \frac{\partial \mathbf{S}^\ell}{\partial \mathbf{W}^Q} \right\|_F^2 = \frac{\sigma_v^2 \sigma_k^2 d^2}{n^2} \cdot \mathbb{E} \left[ \|\mathbf{X}^\ell\|_F^2 \cdot \|(\mathbf{X}^\ell)^\top \mathbf{X}^\ell - n \bar{\boldsymbol{x}}^\ell (\bar{\boldsymbol{x}}^\ell)^\top \|_F^2 \right].
$$

The mean token simplifies to $\bar{\boldsymbol{x}}^l = \frac{\boldsymbol{x}}{n} \sum_k a_k$ and hence $\left( \bar{\boldsymbol{x}}^\ell (\bar{\boldsymbol{x}}^\ell)^\top \right)_{ij} = \frac{1}{n^2} (\sum_k a_k)^2 x_i x_j$. Similarly, $\left( (\mathbf{X}^\ell)^\top \mathbf{X}^\ell \right)_{ij} = \sum_k a_k^2 x_i x_j$. If furthermore all the coefficients $a_i$ are the same (which corresponds to the rank collapse assumption $\mathbf{X}^\ell = \mathbf{1}_n \boldsymbol{x}^T$ analyzed here), then it is easy to see that $\left( (\mathbf{X}^\ell)^\top \mathbf{X}^\ell \right)_{ij} - n \left( \bar{\boldsymbol{x}}^\ell (\bar{\boldsymbol{x}}^\ell)^\top \right)_{ij} = 0 \ \forall i, j$ and hence $\|(\mathbf{X}^\ell)^\top \mathbf{X}^\ell - n \bar{\boldsymbol{x}}^\ell (\bar{\boldsymbol{x}}^\ell)^\top \|_F^2 = 0$.

$\square$

## A.2 Gradient Analysis of Section 3.3

Throughout this section we assume that between every pair of tokens, the same dimension is a zero-mean Gaussian random variable with the same correlation, meaning that

$$\mathbb{E}\left[\mathbf{X}_{i,j}\mathbf{X}_{i',j'}\right] = \begin{cases} 0 & j \neq j' \text{ (independent dimensions)} \\ \sigma_x^2 & i = i', j = j' \\ \rho\sigma_x^2 & i \neq i', j = j'. \end{cases}$$

As we will deal with the computation of 4-th order moments of correlated Gaussian random variables, we will make use of Isserlis theorem [Isserlis, 1918]:

**Theorem A.1** (Isserlis). *Let $X_1, \ldots X_m$ be $m$ zero-mean Gaussian random variables. Then:*

$$\mathbb{E}[X_1 \cdots X_m] = \begin{cases} \sum_{p \in P_m^2} \prod_{(i,j) \in p} \mathbb{E}[X_i X_j] & m \text{ even} \\ 0 & m \text{ odd} \end{cases} \tag{41}$$

*where $P_m^2$ is the set of all the possible pairings of the indexes $1, \ldots, m$.*

In particular, we will only need the 4-th order term, which reads:

$$\mathbb{E}[X_1 X_2 X_3 X_4] = \mathbb{E}[X_1 X_2]\mathbb{E}[X_3 X_4] + \mathbb{E}[X_1 X_3]\mathbb{E}[X_2 X_4] + \mathbb{E}[X_1 X_4]\mathbb{E}[X_2 X_3].$$

Now we can prove Eq. (17) , which we re-state here:

$$\mathbb{E}\left\|\frac{\partial \mathbf{S}}{\partial \mathbf{W}^V}\right\|_F^2 = \sigma_x^2 d^2 \left(1 + \rho(n-1)\right).$$

Also, from Eq. (6) we have that

$$\mathbb{E}\left\|\frac{\partial \mathbf{S}^\ell}{\partial \mathbf{W}^{V,\ell}}\right\|_F^2 = dn\mathbb{E}\|\bar{\boldsymbol{x}}^\ell\|^2.$$

Now,

$$\mathbb{E}\|\bar{\boldsymbol{x}}^\ell\|^2 = \mathbb{E}\left(\sum_{i=1}^d (\bar{\boldsymbol{x}}_i^\ell)^2\right).$$

Each $\bar{\boldsymbol{x}}_i^\ell = \frac{1}{n}\sum_{k=1}^n \mathbf{X}_{ki}^\ell$ is equally distributed with mean

$$\mathbb{E}[\bar{\boldsymbol{x}}_i^\ell] = \mathbb{E}\left[\frac{1}{n}\sum_{k=1}^n \mathbf{X}_{ki}^\ell\right] = 0$$

and variance

$$\text{Var}[\bar{\boldsymbol{x}}_i^\ell] = \text{Var}\left[\frac{1}{n}\sum_{k=1}^n \mathbf{X}_{ki}^\ell\right] = \frac{1}{n^2}\left(n\sigma_x^2 + n(n-1)\rho\sigma_x^2\right) = \frac{1}{n}\sigma_x^2(1 + \rho(n-1)).$$

Finally we get

$$\mathbb{E}\left\|\frac{\partial \mathbf{S}^\ell}{\partial \mathbf{W}^{V,\ell}}\right\|_F^2 = \sigma_x^2 d^2(1 + \rho(n-1)).$$

We know prove Eq. (18), which reads:

$$\mathbb{E}\left\|\frac{\partial \mathbf{S}}{\partial \mathbf{W}^Q}\right\|_F^2 = \sigma_x^6 \frac{(n-1)}{n}(1-\rho)^2 d(n+d).$$

For the queries (and the keys respectively), recall from Eq. (7) that

$$\mathbb{E}\left\|\frac{\partial \mathbf{S}^\ell}{\partial \mathbf{W}^{Q,\ell}}\right\|_F^2 = \frac{\sigma_v^2 \sigma_k^2 d^2}{dn^2} \cdot \mathbb{E}\left[\|\mathbf{X}^\ell\|_F^2 \cdot \|(\mathbf{X}^\ell)^\top \mathbf{X}^\ell - n\bar{\boldsymbol{x}}^\ell (\bar{\boldsymbol{x}}^\ell)^\top\|_F^2\right].$$

To proceed, we drop the superscript $\ell$ and we make the additional assumption that $\|\mathbf{X}\|_F^2$ is uncorrelated from the correlation magnitude $\|\mathbf{X}^\top \mathbf{L}\mathbf{X}\|_F^2 = \|\mathbf{X}^\top \mathbf{X} - n\bar{\boldsymbol{x}}\bar{\boldsymbol{x}}^\top\|_F^2$.

Let us proceed with an expansion:

$$\mathbb{E}\left[\|\mathbf{X}^\top \mathbf{L}\mathbf{X}\|_F^2\right] = \sum_{i,j=1}^d \mathbb{E}\left[\left(\sum_{a,b=1}^n \mathbf{X}_{ai}\mathbf{L}_{ab}\mathbf{X}_{bj}\right)^2\right]$$

$$= \sum_{i,j=1}^d \sum_{a,b,a',b'=1}^n \mathbf{L}_{ab}\mathbf{L}_{a'b'}\mathbb{E}\left[\mathbf{X}_{ai}\mathbf{X}_{a'i}\mathbf{X}_{bj}\mathbf{X}_{b'j}\right].$$

Now, we have 2 cases: if $i \neq j$, which gives $d(d-1)$ equal terms, we need to compute

$$A := \sum_{a,b,a',b'=1}^n \mathbf{L}_{ab}\mathbf{L}_{a'b'}\mathbb{E}\left[\mathbf{X}_{ai}\mathbf{X}_{a'i}\right] \cdot \mathbb{E}\left[\mathbf{X}_{bj}\mathbf{X}_{b'j}\right],$$

where $(i,j)$ is any tuple with $i \neq j$ and we used uncorrelation of different dimensions. Otherwise, we get $d$ each equal to

$$B := \sum_{a,b,a',b'=1}^n \mathbf{L}_{ab}\mathbf{L}_{a'b'}\mathbb{E}\left[\mathbf{X}_{ai}\mathbf{X}_{a'i}\mathbf{X}_{bi}\mathbf{X}_{b'i}\right],$$

where $i$ is any index.

**Term A.** Note that

$$\mathbb{E}\left[\mathbf{X}_{ai}\mathbf{X}_{a'i}\right] \cdot \mathbb{E}\left[\mathbf{X}_{bj}\mathbf{X}_{b'j}\right] = \begin{cases} \sigma_x^4 & a = a', b = b', & n^2 \text{ terms} \\ \rho\sigma_x^4 & a = a', b \neq b', & n^2(n-1) \text{ terms} \\ \rho\sigma_x^4 & a \neq a', b = b', & n^2(n-1) \text{ terms} \\ \rho^2\sigma_x^4 & a \neq a', b \neq b', & n^2(n-1)^2 \text{ terms} \end{cases}.$$

So basically $A$ is the sum of 3 terms:

$$A_1 := \sigma_x^4 \sum_{a,b} \mathbf{L}_{ab}^2 = (n-1)\sigma_x^4$$

$$A_2 := 2\rho\sigma_x^4 \sum_{a,b}\sum_{b'\neq b} \mathbf{L}_{ab}\mathbf{L}_{ab'} = -2(n-1)\rho\sigma_x^4$$

$$A_3 := \rho^2\sigma_x^4 \sum_{a,b}\sum_{a'\neq a}\sum_{b'\neq b} \mathbf{L}_{ab}\mathbf{L}_{a'b'} = \rho^2\sigma_x^4(n-1),$$

where we leveraged the following direct calculations:

$$A_1 = \sigma_x^4 \sum_{a,b} \mathbf{L}_{ab}^2$$

$$= \sigma_x^4 \left(n\left(\frac{n-1}{n}\right)^2 + (n-1)n\frac{1}{n^2}\right)$$

$$= \sigma_x^4 \frac{(n-1)^2 + (n-1)}{n}$$

$$= \sigma_x^4(n-1).$$

Next, we compute

$$A_2 = 2\rho\sigma_x^4 \sum_{a,b} \mathbf{L}_{ab} \sum_{b' \neq b} \mathbf{L}_{ab'}$$

$$= 2\rho\sigma_x^4 \left( \sum_a \mathbf{L}_{a,a} \sum_{b' \neq b} \mathbf{L}_{ab'} + \sum_a \sum_{b \neq a} \mathbf{L}_{ab} \sum_{b' \neq b} \mathbf{L}_{ab'} \right)$$

$$= 2\rho\sigma_x^4 \left( \sum_a \frac{n-1}{n} \left[ -(n-1)\frac{1}{n} \right] + \sum_a \sum_{b \neq a} \mathbf{L}_{ab} \left[ \frac{n-1}{n} - \frac{n-2}{n} \right] \right)$$

$$= 2\rho\sigma_x^4 \left( -\frac{(n-1)^2}{n} - \frac{1}{n}(n-1)n\frac{1}{n} \right)$$

$$= -2\rho\sigma_x^4(n-1).$$

Finally, similar computations also lead to the last term. To follow the calculations, we invite the reader to draw the matrix $\mathbf{L}$ and to hide the columns over which summations are not performed:

$$A_3 = \rho^2\sigma_x^4 \sum_{a,b} \mathbf{L}_{ab} \sum_{a' \neq a} \sum_{b' \neq b} \mathbf{L}_{a'b'}$$

$$= \rho^2\sigma_x^4 \left( \sum_a \mathbf{L}_{aa} \sum_{a' \neq a} \sum_{b' \neq a} \mathbf{L}_{a'b'} + \sum_a \sum_{b \neq a} \mathbf{L}_{ab} \sum_{a' \neq a} \sum_{b' \neq b} \mathbf{L}_{a'b'} \right)$$

$$= \rho^2\sigma_x^4 \left( \sum_a \mathbf{L}_{aa}(1 - \frac{1}{n}) + \sum_a \sum_{b \neq a} \mathbf{L}_{ab}(-1\frac{1}{n}) \right)$$

$$= \rho^2\sigma_x^4 \left( n(1 - \frac{1}{n})(1 - \frac{1}{n}) + n(n-1)(-\frac{1}{n})(-\frac{1}{n}) \right)$$

$$= \rho^2\sigma_x^4(n-1).$$

All in all, we get:

$$\mathrm{A} = (n-1)(1-\rho)^2\sigma_x^4.$$

**Term B.** We make use of Isserlis theorem, stating that:

$$\mathbb{E}\left[\mathbf{X}_{ai}\mathbf{X}_{a'i}\mathbf{X}_{bi}\mathbf{X}_{b'i}\right] = \underbrace{\mathbb{E}\mathbf{X}_{ai}\mathbf{X}_{a'i}\mathbb{E}\mathbf{X}_{bi}\mathbf{X}_{b'i}}_{Q_1} + \underbrace{\mathbb{E}\mathbf{X}_{ai}\mathbf{X}_{bi}\mathbb{E}\mathbf{X}_{a'i}\mathbf{X}_{b'i}}_{Q_2} + \underbrace{\mathbb{E}\mathbf{X}_{ai}\mathbf{X}_{b'i}\mathbb{E}\mathbf{X}_{a'i}\mathbf{X}_{bi}}_{Q_3}.$$

By using our independence assumptions, we get:

$$Q_1 = \sigma_x^4(\delta_{aa'} + \rho\delta_{a \neq a'})(\delta_{bb'} + \rho\delta_{b \neq b'}) = \sigma_x^4(\delta_{aa'}\delta_{bb'} + \rho\delta_{aa'}\delta_{b \neq b'} + \rho\delta_{a \neq a'}\delta_{bb'} + \rho^2\delta_{a \neq a'}\delta_{b \neq b'}).$$

Similarly for $Q_2$ and $Q_3$:

$$Q_2 = \sigma_x^4(\delta_{ab}\delta_{a'b'} + \rho\delta_{ab}\delta_{a' \neq b'} + \rho\delta_{a \neq b}\delta_{a'b'} + \rho^2\delta_{a \neq b}\delta_{a' \neq b'})$$

and

$$Q_3 = \sigma_x^4(\delta_{ab'}\delta_{a'b} + \rho\delta_{ab'}\delta_{a' \neq b} + \rho\delta_{a \neq b'}\delta_{a'b} + \rho^2\delta_{a \neq b'}\delta_{a' \neq b}).$$

Hence,

$$B = \sum_{a,b,a',b'=1}^n \mathbf{L}_{ab}\mathbf{L}_{a'b'}\mathbb{E}\left[\mathbf{X}_{ai}\mathbf{X}_{a'i}\mathbf{X}_{bi}\mathbf{X}_{b'i}\right] = \sum_{a,b,a',b'=1}^n \mathbf{L}_{ab}\mathbf{L}_{a'b'}(Q_1 + Q_2 + Q_3).$$

Let's study it term by term. We will also use $\mathbf{L}_{ab} = (\delta_{ab} - \frac{1}{n})$, and so $\mathbf{L}_{ab}\mathbf{L}_{a'b'} = (\delta_{ab}\delta_{a'b'} - \frac{\delta_{ab}}{n} - \frac{\delta_{a'b'}}{n} + \frac{1}{n^2})$.

**First term**: we have that $\sigma_x^4 \sum_{aa'b'b'} \mathbf{L}_{ab}\mathbf{L}_{a'b'}Q_1$ which is equal to (omitting the constant $\sigma_x^4$):

$$= \sum_{a,a',b,b'} (\delta_{ab}\delta_{a'b'} - \frac{\delta_{ab}}{n} - \frac{\delta_{a'b'}}{n} + \frac{1}{n^2})(\delta_{aa'}\delta_{bb'} + \rho\delta_{aa'}\delta_{b\neq b'} + \rho\delta_{a\neq a'}\delta_{bb'} + \rho^2\delta_{a\neq a'}\delta_{b\neq b'})$$

$$= \rho^2\left(n(n-1) - 2(n-1)(n-1) + (n-1)^2\right) + \rho(-4(n-1) + 2(n-1)) + n - 2 + 1$$

$$= \rho^2(n-1)\left(n - 2(n-1) + (n-1)\right) - 2\rho(n-1) + (n-1)$$

$$= \rho^2(n-1) - 2\rho(n-1) + (n-1).$$

**Second term**: we have that $\sigma_x^4 \sum_{aa'b'b'} \mathbf{L}_{ab}\mathbf{L}_{a'b'}Q_2$ which is equal to (omitting the constant $\sigma_x^4$):

$$= \sum_{a,a',b,b'} (\delta_{ab}\delta_{a'b'} - \frac{\delta_{ab}}{n} - \frac{\delta_{a'b'}}{n} + \frac{1}{n^2})(\delta_{ab}\delta_{a'b'} + \rho\delta_{ab}\delta_{a'\neq b'} + \rho\delta_{a\neq b}\delta_{a'b'} + \rho^2\delta_{a\neq b}\delta_{a'\neq b'})$$

$$= \rho^2(n-1)^2 + \rho(-2n(n-1) + 2(n-1)) + n^2 - 2n + 1$$

$$= \rho^2(n-1)^2 - 2\rho(n-1)^2 + (n-1)^2.$$

**Third term**: we have that $\sigma_x^4 \sum_{aa'b'b'} \mathbf{L}_{ab}\mathbf{L}_{a'b'}Q_3$ which is equal to (omitting the constant $\sigma_x^4$):

$$= \sum_{a,a',b,b'} (\delta_{ab}\delta_{a'b'} - \frac{\delta_{ab}}{n} - \frac{\delta_{a'b'}}{n} + \frac{1}{n^2})(\delta_{ab'}\delta_{a'b} + \rho\delta_{ab'}\delta_{a'\neq b} + \rho\delta_{a\neq b'}\delta_{a'b} + \rho^2\delta_{a\neq b'}\delta_{a'\neq b})$$

$$= \rho^2\left(n(n-1) - 2(n-1)(n-1) + (n-1)^2\right) + \rho(-4(n-1) + 2(n-1)) + n - 2 + 1$$

$$= \rho^2(n-1) - 2\rho(n-1) + (n-1).$$

Summing all the three terms, we get:

$$B = \sigma_x^4(n-1)\left[\rho^2(n+1) - 2(n+1)\rho + (n+1)\right] = \sigma_x^4(n-1)(n+1)(1-\rho)^2.$$

**Plugging in the values of A and B** we get:

$$\mathbb{E}[\|\mathbf{X}^\top\mathbf{L}\mathbf{X}\|_F^2] = d \cdot B + d(d-1) \cdot A = \sigma_x^4(1-\rho)^2 d(n-1)(n+d),$$

and finally assuming Xavier initialization

$$\mathbb{E}\left\|\frac{\partial\mathbf{S}^\ell}{\partial\mathbf{W}^{Q,\ell}}\right\|_F^2 = \frac{\sigma_v^2\sigma_k^2 d^2}{dn^2} \cdot \mathbb{E}\left[\|\mathbf{X}^\ell\|_F^2 \cdot \|(\mathbf{X}^\ell)^\top\mathbf{X}^\ell - n\bar{\mathbf{x}}^\ell(\bar{\mathbf{x}}^\ell)^\top\|_F^2\right]$$

$$= \sigma_x^6\frac{n-1}{n}(1-\rho)^2 d(n+d).$$

## A.3  Forward Pass: Proofs of Lemma 3.2 and 3.3

First, we characterize the evolution of the correlations between tokens $\mathbf{X}_k, \mathbf{X}_{k'}$ with depth, under the assumptions of Theorem 3.3, namely uniform-attention assumption, and the adoption of a linear activation.

**Lemma A.4** (Expectation of Linear Layers). *Let $\mathbf{D} = \mathbf{X}\mathbf{W}$, where $\mathbf{W} \in \mathbb{R}^{d\times d}$ is a random matrix with i.i.d random entries with variance $\sigma^2 = \frac{1}{d}$ and $\mathbf{X} \in \mathbb{R}^{n\times d}$ is a fixed matrix:*

$$\mathbb{E}[\mathbf{D}_{kj}\mathbf{D}_{k'j}] = \frac{1}{d}\langle\mathbf{X}_k, \mathbf{X}_{k'}\rangle$$

Note that by summing over the indexes, Lemma A.4 implies:

$$\mathbb{E}\|\mathbf{D}\|_F^2 = \mathbb{E}\|\mathbf{X}\|_F^2$$
$$\mathbb{E}C(\mathbf{D}) = \mathbb{E}C(\mathbf{X}).$$

*Proof.*

$$\mathbb{E}[\mathbf{D}_{kj}\mathbf{D}_{k'j}] = \sum_{zz'}\mathbf{X}_{kz}\mathbf{X}_{k'z'}\mathbb{E}[\mathbf{W}_{zj}\mathbf{W}_{z'j}] = \sigma^2\sum_z\mathbf{X}_{kz}\mathbf{X}_{k'z} = \frac{1}{d}\langle\mathbf{X}_k, \mathbf{X}_{k'}\rangle.$$

$\square$

**Lemma A.5** (Expectation of skip connection). *Let* $\mathbf{A}, \mathbf{B} \in \mathbb{R}^{p \times q}$. *Let* $\mathbf{D} := \alpha \mathbf{A} + \mathbf{B}$ *with* $\mathbb{E}[\mathbf{A}|\mathbf{B}] = \mathbf{0}$ *and* $\alpha \in \mathbb{R}$. *Then:*

$$\mathbb{E}\left[\mathbf{D}_{ij}\mathbf{D}_{i'j}\right] = \alpha^2 \mathbb{E}[\mathbf{A}_{ij}\mathbf{A}_{i'j}] + \mathbb{E}[\mathbf{B}_{ij}\mathbf{B}_{ij'}] \tag{42}$$

*holds for all* $i, i' \in [p], j \in [q]$.

Note that by summing over the indexes, Lemma A.5 implies:

$$\mathbb{E}\|\mathbf{D}\|_F^2 = \alpha^2 \mathbb{E}\|\mathbf{A}\|_F^2 + \mathbb{E}\|\mathbf{B}\|_F^2$$
$$\mathbb{E}C(\mathbf{D}) = \alpha^2 \mathbb{E}C(\mathbf{A}) + \mathbb{E}C(\mathbf{B}).$$

*Proof.*

$$\mathbb{E}[\mathbf{D}_{ij}\mathbf{D}_{i'j}] = \mathbb{E}\left[(\alpha\mathbf{A}_{ij} + \mathbf{B}_{ij})(\alpha\mathbf{A}_{i'j} + \mathbf{B}_{i'j})\right]$$
$$= \mathbb{E}\left[\alpha^2 \mathbf{A}_{ij}\mathbf{A}_{i'j} + \alpha\mathbf{A}_{ij}\mathbf{B}_{i'j} + \alpha\mathbf{A}_{i'j}\mathbf{B}_{ij} + \mathbf{B}_{ij}\mathbf{B}_{i'j}\right]$$
$$= \alpha^2 \mathbb{E}\left[\mathbf{A}_{kj}\mathbf{A}_{i'j}\right] + \mathbb{E}\left[\mathbf{B}_{ij}\mathbf{B}_{i',j}\right],$$

where using iterated expectations $\alpha\mathbb{E}[\mathbf{A}_{i'j}\mathbf{B}_{ij}] = \alpha\mathbb{E}[\mathbb{E}[\mathbf{A}_{i'j}|\mathbf{B}]\mathbf{B}_{ij}]] = 0$ and identically $\alpha\mathbb{E}[\mathbf{A}_{ij}\mathbf{B}_{i'j}] = 0$. $\square$

**Lemma A.6** (Expectation of Attention Layers). *Under the uniform-attention assumption:*

$$\mathbb{E}[\mathbf{S}_{kj}\mathbf{S}_{k'j}] = \frac{1}{d_v n^2}\mathbb{E}C(\mathbf{X}).$$

In this case, by summing over the indexes we have that:

$$\mathbb{E}\|\mathbf{S}\|_F^2 = \frac{\mathbb{E}C(\mathbf{X})}{n}$$
$$\mathbb{E}C(\mathbf{S}) = \mathbb{E}C(\mathbf{X}).$$

*Proof.* Note that under the uniform-attention assumption:

$$\mathbf{S}_{kj} = \frac{1}{n}\left(\mathbf{1}_{n\times n}\mathbf{X}\mathbf{W}^V\right)_{kj} = \frac{1}{n}\sum_{zi}\mathbf{X}_{zi}\mathbf{W}_{ij}^V.$$

Hence, using the fact that the weights are i.i.d with variance $\sigma_v^2 = \frac{1}{d_v}$:

$$\mathbb{E}[\mathbf{S}_{kj}\mathbf{S}_{k'j}] = \frac{\sigma_v^2}{n^2}\sum_{z,z'}\sum_i \mathbb{E}[\mathbf{X}_{zi}\mathbf{X}_{z'i}] = \frac{1}{d_v n^2}\sum_{k,k'}\langle\mathbf{X}_z, \mathbf{X}_{z'}\rangle = \frac{1}{d_v n^2}\mathbb{E}C(\mathbf{X}).$$

$\square$

**Lemma 3.2** (Propagation of inner products). *Let* $C(\mathbf{X}^\ell) = \sum_{k,k'}\langle\mathbf{X}_k^\ell, \mathbf{X}_{k'}^\ell\rangle$ *and* $\mathbf{X}$ *the input sequence. Under the Assumption 3.1 and if* $\sigma$ *is the linear activation function, we have that:*

$$\mathbb{E}\left[C(\mathbf{X}^L)\right] = (\alpha_2^2 + 1)^L(\alpha_1^2 + 1)^L C(\mathbf{X}). \tag{10}$$

*hence, under the depth scaling for the residual block parameters* $\alpha_1^2 = \frac{\tilde{\alpha}_1}{L}, \alpha_2^2 = \frac{\tilde{\alpha}_2}{L}$ *with* $\tilde{\alpha}_1, \tilde{\alpha}_2 \in \mathbb{R}$ *independent of* $L$, *we have that:*

$$\lim_{L\to\infty}\mathbb{E}[C(\mathbf{X}^L)] = e^{\tilde{\alpha}_1 + \tilde{\alpha}_2}C(\mathbf{X}). \tag{11}$$

*Proof.* First, note that for the residual blocks we have that $\mathbb{E}[Y_{kj}^\ell|Z_{k'j}^\ell] = 0$ due to the independence assumption on the feedforward weights, and similarly $\mathbb{E}[S_{kj}^\ell|X_{k'j}^\ell] = 0$. Hence, we can use Lemma

A.5 in both the skip connections of the Transformer architecture. Therefore, using Lemma A.5 (skip), Lemma A.4 (linear) and Lemma A.6 (attention):

$$
\begin{aligned}
\mathbb{E}[C(\mathbf{X}^{\ell+1})] \\
&\overset{\text{skip}}{=} \alpha_2^2 \mathbb{E}C(\mathbf{Y}^\ell) + \mathbb{E}C(\mathbf{Z}^\ell) \\
&\overset{\text{linear}}{=} \alpha_2^2 \mathbb{E}C(\mathbf{Z}^\ell) + \mathbb{E}C(\mathbf{Z}^\ell) \\
&= (\alpha_2^2 + 1)\mathbb{E}C(\mathbf{Z}^\ell) \\
&\overset{\text{skip}}{=} (\alpha_2^2 + 1)\left(\alpha_1^2 \mathbb{E}C(\mathbf{S}^\ell) + \mathbb{E}C(\mathbf{X}^\ell)\right) \\
&\overset{\text{attention}}{=} (\alpha_2^2 + 1)(\alpha_1^2 + 1)\mathbb{E}[C(\mathbf{X}^\ell)] \\
&\overset{\text{unroll recurs.}}{=} (\alpha_2^2 + 1)^{\ell+1}(\alpha_1^2 + 1)^{\ell+1} C(\mathbf{X}),
\end{aligned}
$$

where in the last step we have unrolled the recursion until the input layer.

For the limit as $L \to \infty$, simply note that:

$$
\lim_{L \to \infty} \left(\frac{\tilde{\alpha}_i}{L} + 1\right)^L = e^{\tilde{\alpha}_i},
$$

with $i \in \{1, 2\}$. $\square$

Now we are ready to re-state and prove Lemma 3.3.

**Lemma 3.3** (Propagation of the norm). *Let $\mathbf{X}^L$ be the representations of the input sequence at the final layer. Under the assumptions of Lemma 3.2, we have that:*

$$
\mathbb{E}\left\|\mathbf{X}^L\right\|_F^2 = n(\alpha_2^2 + 1)^L \alpha_1^2 \sum_{k=0}^{L-1} (\alpha_1^2 + 1)^k \left\|\bar{\boldsymbol{x}}\right\|^2 + (\alpha_2^2 + 1)^L \|\mathbf{X}\|_F^2, \tag{12}
$$

*hence, under the depth scaling for the residual block parameters $\alpha_1^2 = \frac{\tilde{\alpha}_1}{L}, \alpha_2^2 = \frac{\tilde{\alpha}_2}{L}$ with $\tilde{\alpha}_1, \tilde{\alpha}_2 \in \mathbb{R}$ independent of L, we have that:*

$$
\lim_{L \to \infty} \mathbb{E}\left\|\mathbf{X}^L\right\|_F^2 = ne^{\tilde{\alpha}_2}(e^{\tilde{\alpha}_1} - 1)\left\|\bar{\boldsymbol{x}}\right\|^2 + e^{\tilde{\alpha}_2}\|\mathbf{X}\|_F^2. \tag{13}
$$

*Proof.* The proof is in the same spirit as Lemma 3.2 but slightly more involved. Again, using Lemma A.5 in both the skip connections of the Transfomer architecture. Therefore, using Lemma A.5 (skip), Lemma A.4 (linear) and Lemma A.6 (attention):

$$
\begin{aligned}
\mathbb{E}[||\mathbf{X}^{\ell+1}||_F^2] &\overset{\text{skip}}{=} \alpha_2^2 \mathbb{E}||\mathbf{Y}^\ell||_F^2 + \mathbb{E}||\mathbf{Z}^\ell||_F^2 \\
&\overset{\text{linear}}{=} (\alpha_2^2 + 1)\mathbb{E}||\mathbf{Z}^\ell||_F^2 \\
&\overset{\text{skip}}{=} (\alpha_2^2 + 1)\left(\alpha_1^2 \mathbb{E}[||\mathbf{S}^\ell||_F^2] + \mathbb{E}[||\mathbf{X}^\ell||_F^2]\right) \\
&\overset{\text{softmax}}{=} (\alpha_2^2 + 1)\left(\frac{\alpha_1^2}{n}\mathbb{E}[C(\mathbf{X}^\ell)] + \mathbb{E}[||\mathbf{X}^\ell||_F^2]\right) \\
&= (\alpha_2^2 + 1)\frac{\alpha_1^2}{n}\mathbb{E}[C(\mathbf{X}^\ell)] + (\alpha_2^2 + 1)\mathbb{E}[||\mathbf{X}^\ell||_F^2] \\
&\overset{\text{unroll } C(\mathbf{X}^\ell)}{=} (\alpha_2^2 + 1)^{\ell+1}\alpha_1^2(\alpha_1^2 + 1)^\ell \frac{C(\mathbf{X})}{n} + (\alpha_2^2 + 1)\mathbb{E}[||\mathbf{X}^\ell||_F^2] \\
&\overset{\text{unroll } ||\mathbf{X}^\ell||_F^2}{=} (\alpha_2^2 + 1)^{\ell+1}\alpha_1^2 \frac{C(\mathbf{X})}{n}\sum_{k=0}^{\ell}(\alpha_1^2 + 1)^k + (\alpha_2^2 + 1)^{\ell+1}||\mathbf{X}||_F^2,
\end{aligned}
$$

where in the second to last step we have used Lemma 3.2 and in the last step we have unrolled the recursion for $\left\|\mathbf{X}^\ell\right\|_F^2$ until the input layer.

For the second part, we now show that for a network of $L$ layers, the choice $\alpha_1^2 = \frac{\tilde{\alpha}_1}{L}$ and $\alpha_2^2 = \frac{\tilde{\alpha}_2}{L}$ stabilizes the norm of the activations in the forward pass. Using the product law for the limits, we can study the converges of $(\alpha_2^2 + 1)^\ell$ and $\alpha_1 \sum_{k=0}^{\ell}(\alpha_1^2 + 1)^k$ separately.

Let $i \in \{1, 2\}$. For the latter term we have that:

$$\lim_{L \to \infty} \frac{\tilde{\alpha}_i}{L} \sum_{l=0}^{L-1} \left(1 + \frac{\tilde{\alpha}_i}{L}\right)^\ell = \lim_{L \to \infty} \frac{\tilde{\alpha}_i}{L} \frac{1 - \left(1 + \frac{\tilde{\alpha}_i}{L}\right)^\ell}{1 - 1 - \frac{\tilde{\alpha}_i}{L}}$$

$$= \lim_{L \to \infty} -1 + \left(1 + \frac{\tilde{\alpha}_i}{L}\right)^\ell$$

$$= e^{\tilde{\alpha}_i} - 1,$$

while for the former term we have that $\lim_{L \to \infty}(\frac{\tilde{\alpha}_i}{L} + 1)^\ell = e^{\tilde{\alpha}_i}$. Hence, the norm of the representations converges to:

$$\lim_{L \to \infty} \mathbb{E}[||\mathbf{X}^\ell||_F^2] = e^{\tilde{\alpha}_2}(e^{\tilde{\alpha}_1} - 1)\frac{C(\mathbf{X})}{n} + e^{\tilde{\alpha}_2}||\mathbf{X}||_F^2.$$

The final results as stated in the theorem hold because of the following:

**Remark**: note that $C(\mathbf{X}) = \sum_{k,k'} \sum_j \mathbf{X}_{kj}\mathbf{X}_{kj'} = \sum_j (\sum_{k,k'} \mathbf{X}_{kj}\mathbf{X}_{k'j}) = \sum_j (\sum_k \mathbf{X}_{kj})^2 = n^2 \|\bar{\boldsymbol{x}}\|^2$. $\qquad \square$

### A.4 Proof of Theorem 3.3: Correlations are Preserved under Residual Scaling

**Theorem 3.3.** *Let the input tokens have the same norm, i.e.* $\|\mathbf{X}_k\| = \|\boldsymbol{x}\| \;\; \forall k \in [n]$ *for some* $\boldsymbol{x} \in \mathbb{R}^{d_v}$. *Under the depth scaling for the residual block parameters* $\alpha_1^2 = \frac{\tilde{\alpha}_1}{L}, \alpha_2^2 = \frac{\tilde{\alpha}_2}{L}$ *with* $\tilde{\alpha}_1, \tilde{\alpha}_2 \in \mathbb{R}$ *independent of L, we have that:*

$$\lim_{L \to \infty} \rho^\ell = \frac{ne^{\tilde{\alpha}_1}C(\mathbf{X})}{(n-1)[(e^{\tilde{\alpha}_1} - 1)C(\mathbf{X}) + n\|\mathbf{X}\|_F^2]} - \frac{1}{n-1}. \tag{15}$$

*On the other hand, if* $\alpha_1, \alpha_2 \neq 0$ *are some constants independent of L, we have that:*

$$\lim_{L \to \infty} \rho^\ell = 1. \tag{16}$$

*Proof.* Due to the rotational symmetries of the Gaussian random matrices, if the input tokens have the same norm, then the expected norm at layer $\ell \in [L]$ is also the same across the token's representations. Hence, we can write $\mathbb{E}\|\mathbf{X}^\ell\|_F^2 = n\mathbb{E}\|\boldsymbol{x}^\ell\|^2$, where $\|\boldsymbol{x}^\ell\|^2$ is the norm of every token at layer $\ell$. Furthermore, by definition of our correlation coefficient $\rho_{kk'}^l$, we have that $\mathbb{E}\langle \mathbf{X}_k^\ell, \mathbf{X}_{k'}^\ell \rangle = \rho_{kk'}^\ell \mathbb{E}\|\boldsymbol{x}^\ell\|^2$. By summing over the indexes $k, k'$, we can expand the relation as:

$$\underbrace{\sum_{k,k'} \mathbb{E}\langle \mathbf{X}_k^\ell, \mathbf{X}_{k'}^\ell \rangle}_{\mathbb{E}C(\mathbf{X})} = \sum_{k,k'} \rho_{kk'}^\ell \mathbb{E}\|\boldsymbol{x}^\ell\|^2 = (n + \sum_{k \neq k'} \rho_{k,k'}^\ell)\mathbb{E}\|\boldsymbol{x}^\ell\|^2 = \underbrace{n\mathbb{E}\|\boldsymbol{x}^\ell\|^2}_{\mathbb{E}\|\mathbf{X}^\ell\|_F^2}(1 + (n-1)\rho^\ell).$$

By solving for $\rho^\ell$, we have that:

$$\rho^\ell = \frac{\mathbb{E}C(\mathbf{X}^\ell)}{(n-1)\mathbb{E}\|\mathbf{X}^\ell\|^2} - \frac{1}{n-1}.$$

Now we plug in the expressions for $\mathbb{E}C(\mathbf{X}^\ell)$ and $\mathbb{E}\|\mathbf{X}^\ell\|^2$ with the aid of Lemma 3.2 and Lemma 3.3, respectively. Finally, by taking the limits with respect to $L$, we get the desired result. $\qquad \square$

### A.5 Motivation for Assumption 3.1

We motivate here the following assumption, stated in the main paper. This assumption is crucial to compute expectations involving the softmax function.

**Assumption 3.1** (Uniform attention). *We assume that* $\mathbf{A}^\ell = \frac{1}{n}\mathbf{1}_{n \times n}$,

**Theoretical analysis.** We first show that this assumption holds when taking $d_k$ to infinity, keeping $d_v$ fixed.

**Lemma A.7.** *Consider initializing each entry of $\mathbf{W}^Q \in \mathbb{R}^{d_v \times d_k}$ and $\mathbf{W}^K \in \mathbb{R}^{d_v \times d_k}$ independently with variance $\sigma_k^2 = 2/(d_v + d_k)$ — i.e. Glorot initialization [Glorot and Bengio, 2010]. Let $\mathbf{M} = \frac{1}{\sqrt{d_k}} \mathbf{X}^\ell \mathbf{W}^{Q,\ell} \mathbf{W}^{K,\ell^\top} \mathbf{X}^{\ell^\top}$; for any $(i,j) \in [n] \times [n]$ we have*

$$\mathbb{E}[\mathbf{M}_{i,j} \mid \mathbf{X}] = 0, \qquad \mathbb{E}[\mathbf{M}_{i,j}^2 \mid \mathbf{X}] = \sigma_k^4 \cdot \|\mathbf{X}_{i,:}\|^2 \cdot \|\mathbf{X}_{j,:}\|^2. \tag{43}$$

*While keeping $d_v < \infty$ fixed, taking $d_k$ to infinity yields*

$$\mathbb{E}[\mathbf{M}_{i,j}^2 \mid \mathbf{X}] = \mathcal{O}\left(\frac{1}{d_k^2}\right). \tag{44}$$

*In other words, $\mathbf{M}$ converges to $\mathbf{0}_{n \times n}$ in $L^2$ as $d_k \to \infty$.*

*Proof.* First, note that

$$\mathbf{M}_{i,j} = \frac{1}{\sqrt{d_k}} \sum_{a,c=1}^{d_v} \sum_{b=1}^{d_k} \mathbf{X}_{i,a} \mathbf{W}_{a,b}^Q \mathbf{W}_{c,b}^K \mathbf{X}_{j,c}.$$

Since $\mathbf{W}^Q$ is independent from $\mathbf{W}^K$ at initialization, $\mathbb{E}[\mathbf{M}_{i,j} \mid \mathbf{X}] = 0$. Next, we compute

$$\begin{aligned}
\mathbb{E}[\mathbf{M}_{i,j}^2] &= \frac{1}{d_k} \sum_{a,c,a',c'=1}^{d_v} \sum_{b,b'=1}^{d_k} \mathbf{X}_{i,a} \mathbf{X}_{i,a'} \mathbf{X}_{j,c} \mathbf{X}_{j,c'} \mathbb{E}\left[\mathbf{W}_{a,b}^Q \mathbf{W}_{a',b'}^Q \mathbf{W}_{c,b}^K \mathbf{W}_{c',b'}^K\right] \\
&= \frac{1}{d_k} \sum_{a,c,a',c'=1}^{d_v} \sum_{b,b'=1}^{d_k} \mathbf{X}_{i,a} \mathbf{X}_{i,a'} \mathbf{X}_{j,c} \mathbf{X}_{j,c'} \mathbb{E}\left[\mathbf{W}_{a,b}^Q \mathbf{W}_{a',b'}^Q\right] \mathbb{E}\left[\mathbf{W}_{c,b}^K \mathbf{W}_{c',b'}^K\right] \\
&= \frac{\sigma_k^4}{d_k} \sum_{a,c=1}^{d_v} \sum_{b=1}^{d_k} \mathbf{X}_{i,a}^2 \mathbf{X}_{j,c}^2 \\
&= \sigma_k^4 \|\mathbf{X}_{i,:}\|^2 \|\mathbf{X}_{j,:}\|^2.
\end{aligned}$$

This concludes the proof. $\qquad\square$

The following classical result implies almost sure convergence of the softmax matrix as $d_k \to \infty$.

**Lemma A.8** (Borel-Cantelli). *Let $(X_i)$ be a sequence of random variables. If for any $\epsilon > 0$*

$$\sum_{i=0}^{\infty} \mathbb{P}[|X_i - X| > \epsilon] < \infty,$$

*then $X_i$ converges to $X$ almost surely[4].*

**Theorem A.2** (Almost-sure convergence). *Consider initializing each entry of $\mathbf{W}^Q \in \mathbb{R}^{d_v \times d_k}$ and $\mathbf{W}^K \in \mathbb{R}^{d_v \times d_k}$ independently with variance $\sigma_k^2 = 2/(d_v + d_k)$ — i.e. Glorot initialization [Glorot and Bengio, 2010]. Let $d_v < \infty$ be fixed, as $d_k \to \infty$ we have that, for any $\mathbf{X}$,*

$$\mathbf{A} := \mathrm{softmax}\left(\frac{1}{\sqrt{d_k}} \mathbf{X} \mathbf{W}^Q \mathbf{W}^{K^\top} \mathbf{X}^\top\right) \overset{a.s.}{\to} \frac{1}{n} \mathbf{1}_{n \times n}$$

*and*

$$\frac{\partial \mathbf{A}}{\partial \mathbf{M}} \overset{a.s.}{\to} \frac{1}{n} \mathbf{I}_n \otimes \left(\mathbf{I}_n - \frac{1}{n} \mathbf{1}_{n \times n}\right).$$

---

[4]That is, $\lim_{i \to \infty} X_i(\omega) = X(\omega)$ for almost every $\omega \in \Omega$ (i.e. with probability one).

*Proof.* Thanks to Lemma A.7 and Markov Inequality, we have fast convergence in probability: for any fixed $\mathbf{X}$,

$$\mathbb{P}[|\mathbf{M}_{i,j}| > \epsilon] \leq \frac{\mathbb{E}[\mathbf{M}_{i,j}^2]}{\epsilon^2} \leq \frac{C_\epsilon}{d_k^2}.$$

Borel Cantelli then directly yields almost sure convergence of $\mathbf{M}$ to $\mathbf{0}_{n \times n}$ as $d_k \to \infty$. Next, note that both $\mathbf{A}$ and $\frac{\partial \mathbf{A}}{\partial \mathbf{M}}$ are continuous functions of $\mathbf{A}$, hence we can apply standard continuity event-per-event. For almost every $\omega \in \Omega$,

$$\lim_{d_k \to \infty} \mathbf{A}(\mathbf{A}(\omega)) = \mathbf{A}\left(\lim_{d_k \to \infty} \mathbf{A}(\omega)\right) = \mathbf{A}(\mathbf{0}_{n \times n}) = \frac{1}{n}\mathbf{1}_{n \times n}.$$

Hence $\mathbf{A} \to \frac{1}{n}\mathbf{1}_{n \times n}$ almost surely. This can also be seen as a simple application of the continuous mapping theorem. The same reasoning yields almost sure convergence of

$$\frac{\partial \mathbf{A}}{\partial \mathbf{M}} = \text{blockdiag}\left(\text{diag}(\mathbf{A}_{i:}) - \mathbf{A}_{i:}\mathbf{A}_{i:}^\top\right),$$

to the corresponding limiting quantity. $\qquad\square$

**Empirical analysis.** We empirically assess the validity of Assumption 3.1 and of its theoretical justification by performing the following experiments: for a range of increasing values of $d_k$, we compute $\mathbf{A}$ and we calculate $\frac{1}{n^2}||\mathbf{A}^\ell - \frac{1}{n}\mathbf{1}_{n \times n}||_F^2$, i.e. its average (entry-wise) distance from a uniform matrix with entries all equal to $1/n$. For each value of $d_k$, we repeat this calculation 200 times, each time with different random weight matrices. Fig. 7 displays how the $||\mathbf{A} - \frac{1}{n}\mathbf{1}_{n \times n}||_F^2$ averaged over 200 runs, tends to zero with a trend inversely proportional to $d_k^2$, as predicted by our theoretical analysis.

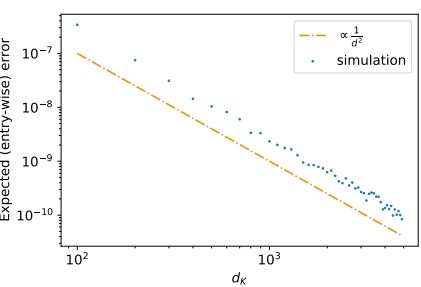

Figure 7: Evolution of $\frac{1}{n^2}||\mathbf{A}^\ell - \frac{1}{n}\mathbf{1}_{n \times n}||_F^2$ as a function of $d_k$ for $d_v$ fixed at 100.

# B  Additional Results

## B.1  On the Roles of the $1/\sqrt{L}$-Scaling of the Residuals and Layer Normalization

We present some additional results on the propagation of the norm and the correlations in Figure 8. In particular, we empirically show that, with an adequate depth-dependent residual scaling, the norm and the correlation are stabilized, even for very deep networks. Furthermore, we demonstrate the propagation of the correlation and the gradient norms for the PRE-LN configuration in Figure 9. As also hinted in the main text, in Figure 4, the increase in correlation with depth for PRE-LN is much less wild. This also results in better stabilized gradients for the queries and keys' parameters. We also observe the opposite trend for the gradients of the values, in relation to the POST-LN case in Figure 3. We speculate that this different dependence, along with the better preserved correlation, is the main reason PRE-LN configured Transformers have been shown to scale better with depth. We plan to investigate this dependence more in future work. We provide the detailed experimental setup for Figures 3 and 9 in Table 2. Finally, as hinted in the main text, replacing the linear activation function with ReLU leads to an even faster increase in correlations (for instance, see the contraction argument in Nachum et al. [2021] below Equation (2)). We empirically evaluate that, by comparing the propagation of correlation (as in Figure 4) also for a network with ReLU activation functions. Results are shown in Figure 10.

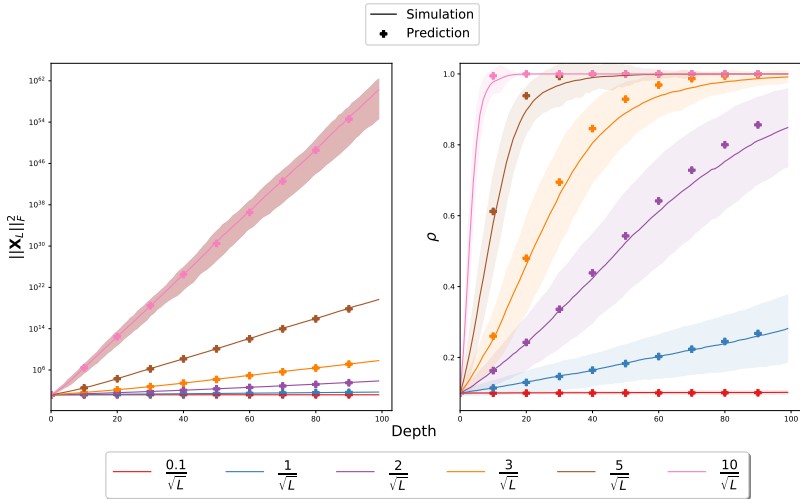

Figure 8: (Left) Propagation of the Frobenius norm of the input sequence; (Right) Propagation of the average token correlation. In both cases, no layer normalization layers are used and linear activations are employed.

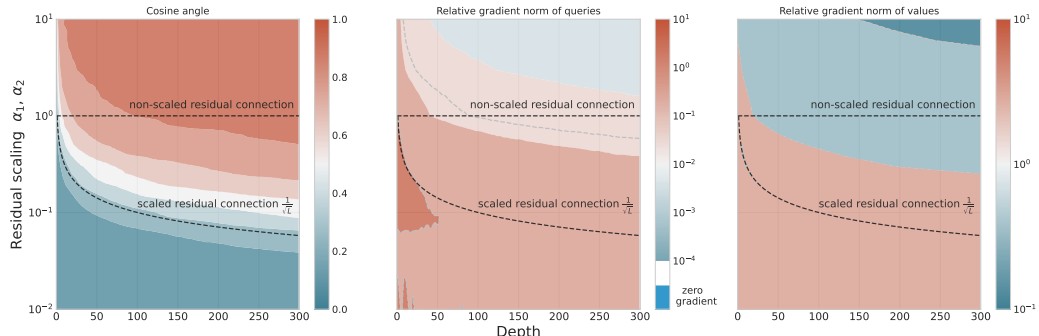

Figure 9: Same as Figure 3 but with a PRE-LN architecture. The correlation at depth 0 originates from the correlations in the randomly initialized tokens' embeddings and positional encodings.

| Hyperparameters | Value |
|---|---|
| Embedding dimension | 32 |
| MLP dimension | 128 |
| Number of heads | 4 |
| Number of tokens | 50 |
| nonlinearity | ReLU |

Table 2: Hyperparameters for Figures 3 and 9. Results are averaged across 50 runs.

## B.2 Further Empirical Assessment of Assumption 3.1

Here, we empirically test the accuracy and limitations of the uniform-attention assumption.

For the empirical verification of Assumption 3.1 in the forward pass analysis, we plot the density of the norm of the representations for only-encoder Transformers of increasing depth. The results are shown in Fig 11. Note that when the standard deviation of the input is set to $1/\sqrt{d}$, then the uniform-attention assumption provide an excellent approximation to the common Xavier-initialization. On the contrary, we observe a deviation when the standard deviation of the input is increased. Also, note how as the depth increases, the distribution becomes more heavy-tailed. This heavy-tailedness

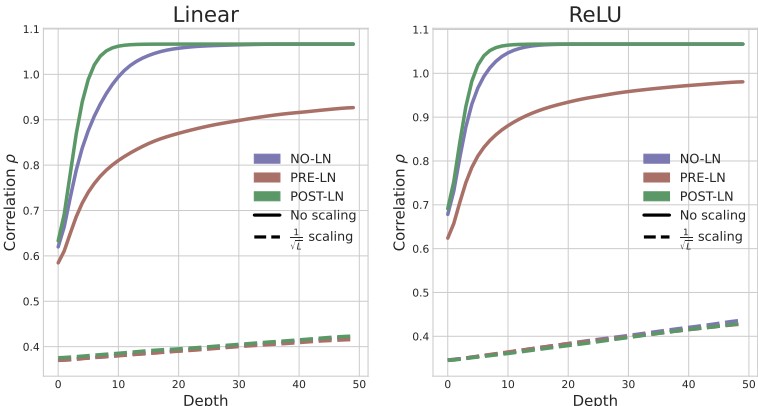

Figure 10: Increase in correlation at initialization for a network with linear (left) and ReLU (right) activation functions.

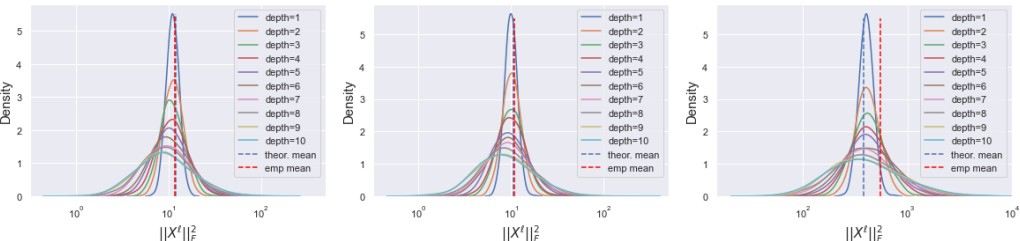

Figure 11: Density plots for $\left\|\mathbf{X}^\ell\right\|_F^2$ for Transformers of depths $L$ from 1 to 10. The input $\mathbf{X}$ contains i.i.d Gaussian entries, simulating an embedding layer. We set $d := d_v = d_q = 30$. The empirical mean at $L = 10$ is highlighted in a vertical dashed red line, while the theoretical mean (Lemma 3.3) is a dashed blue line. The densities are estimated by sampling 1000 times the weights of the network. (Left): we adopt the uniform-attention. The standard deviation of the input is set to $1/\sqrt{d}$. (Center): Same, but removing the uniform-attention assumption. (Right): We remove the uniform-attention assumption, and set the standard deviation of the input to 1.

was recently formally shown for standard MLPs with and without ReLU activation [Noci et al., 2021, Zavatone-Veth and Pehlevan, 2021].

For the verification of the assumption in the backward pass, we additionally show in Fig. 12 how the norm of the gradients w.r.t queries and keys depends on the hidden dimension, the sequence length, the input correlation and the input variance. *Ground-truth* gradients are calculated with automatic differentiation, and they are compared with our theoretical results based on Assumption 3.1. As shown in Fig.12, our theoretical predictions show a very good agreement with the true gradients. Again, we notice that the smaller the values of the input standard deviation the tighter the agreement of the theory with the simulations. Intuitively, a higher input variance causes the argument of the softmax to have a large range of values. This in turn causes a deviation from the uniform distribution (i.e. maximum entropy), towards the distribution of minimum entropy (a Delta Dirac, corresponding to attending to only one token).

### B.3 Empirical Verification of the Gradient Analysis of Section 3.3

Finally, in Figures 13 and 14 we show the dependence of the norm of the gradients for the keys and values based on the parameters of the architecture and the task-specific parameters. Figure 13 illustrates the true dependence and Figure 14 the one expected by the theory based on our assumptions. In short, the main takeaways are the following.

- As the correlation between the tokens increases ($x$-axis in the global plot), the norm of the gradients of the queries quickly diminishes compared to the one of the values.

- The dependence on the variance of the input $\sigma_x^2$ is different ($y$-axis in the global plot), being linear for the values and cubic for the queries. This highlights the importance of a stabilized forward pass and provides another explanation regarding the successful use of layer norm in Transformers.

- The dependence on $n$ ($x$-axis in each subplot) and $d$ ($y$-axis in each subplot) is more complicated, also being a function of the correlation $\rho$ (compare the first column where $\rho = 0$ to the rest).

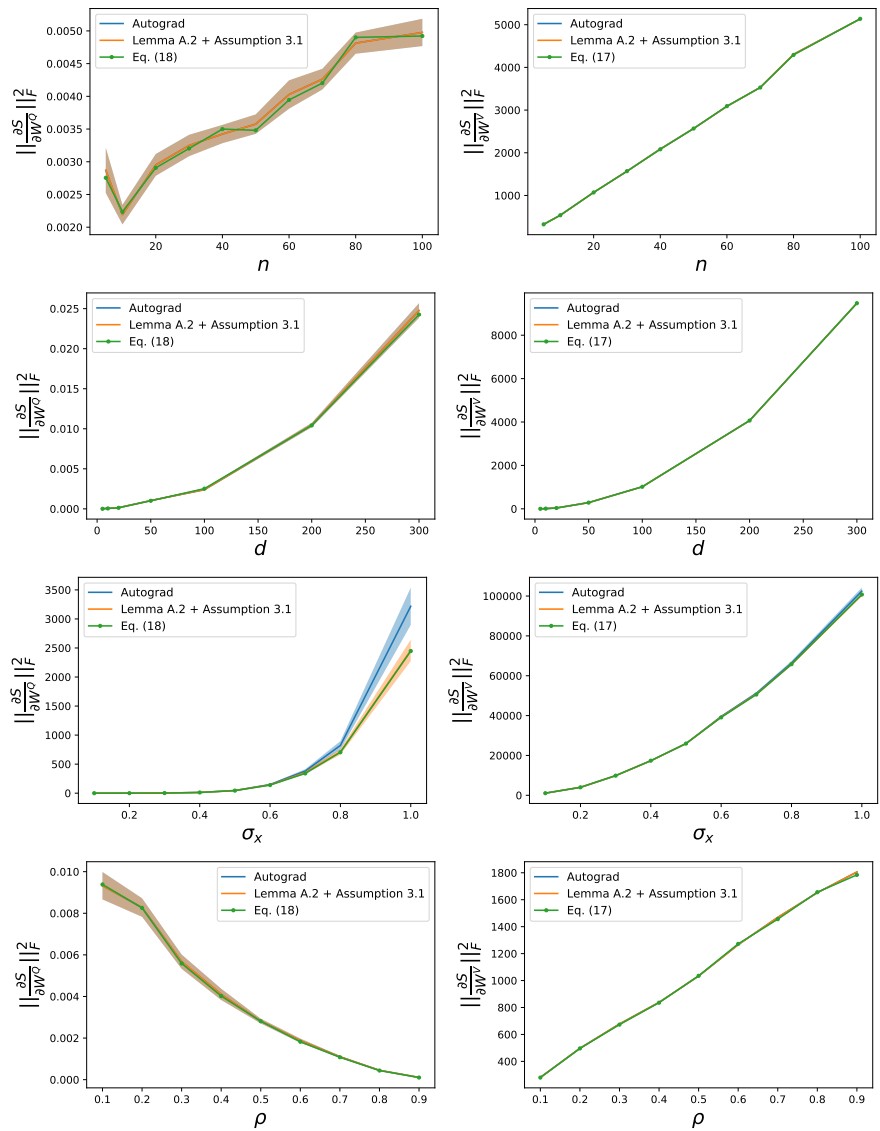

Figure 12: Empirical comparison of our theoretical findings. We sample, as aforementioned, the tokens according to a zero-mean Gaussian distribution, while varying the hidden dimension, sequence length, input correlation and input variance. Results are averaged over 20 runs.

## C  Experimental Setup

Here we provide more details regarding the experimental setup.

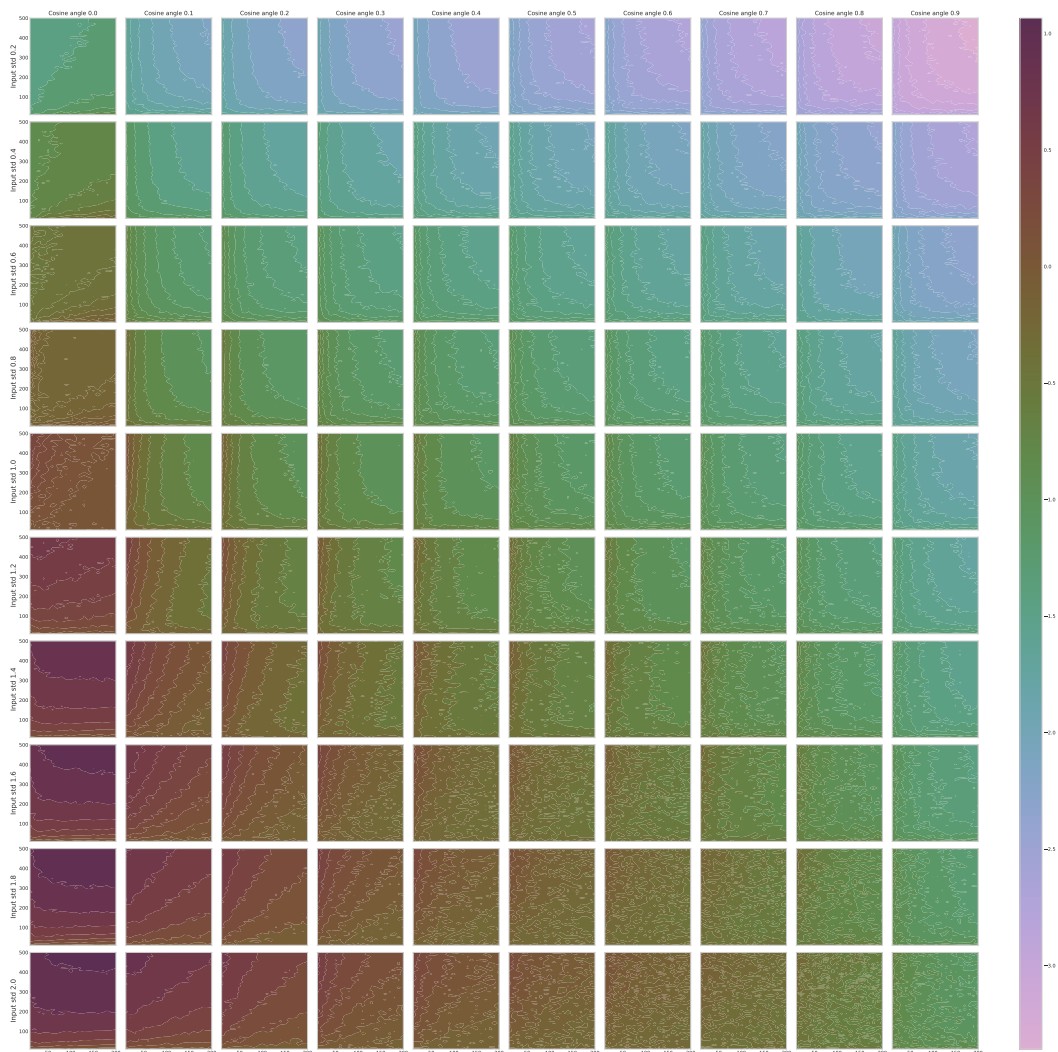

Figure 13: Log ratio of the norm of the gradients for the queries compared to those of the values for varying values of embedding dimension, sequence length, cosine of the tokens angle and standard deviation.

## C.1 Toy Example

In Figure 6, we focus on a toy example where the task is to reverse a sequence of tokens. More specifically, given a sequence of 20 numbers in the range $0 - 9$, we predict the same tokens in the inverted order. We use an embedding layer of size 16, initializes with variance 1, and sinusoidal positional encodings to initially embed the input. We use a 5-layer POST-LN Transformer encoder model, with a single head attention operation and a two-layer feed-forward layer with a ReLU nonlinearity. We use residual scaling in this case equal to $\alpha_1 = \alpha_2 = 1$. We train using Adam with betas parameters $(0.9, 0.999)$, learning rate 0.01 and weight decay 0.

## C.2 Tempering the softmax

The theory devised in Section 3.3 postulates that a different magnitude between the gradients of the queries/keys and the values should likely be observed. Here, we propose a simple remedy that consists in introducing an inverse temperature scaling $\tau$ inside the softmax that modifies the attention

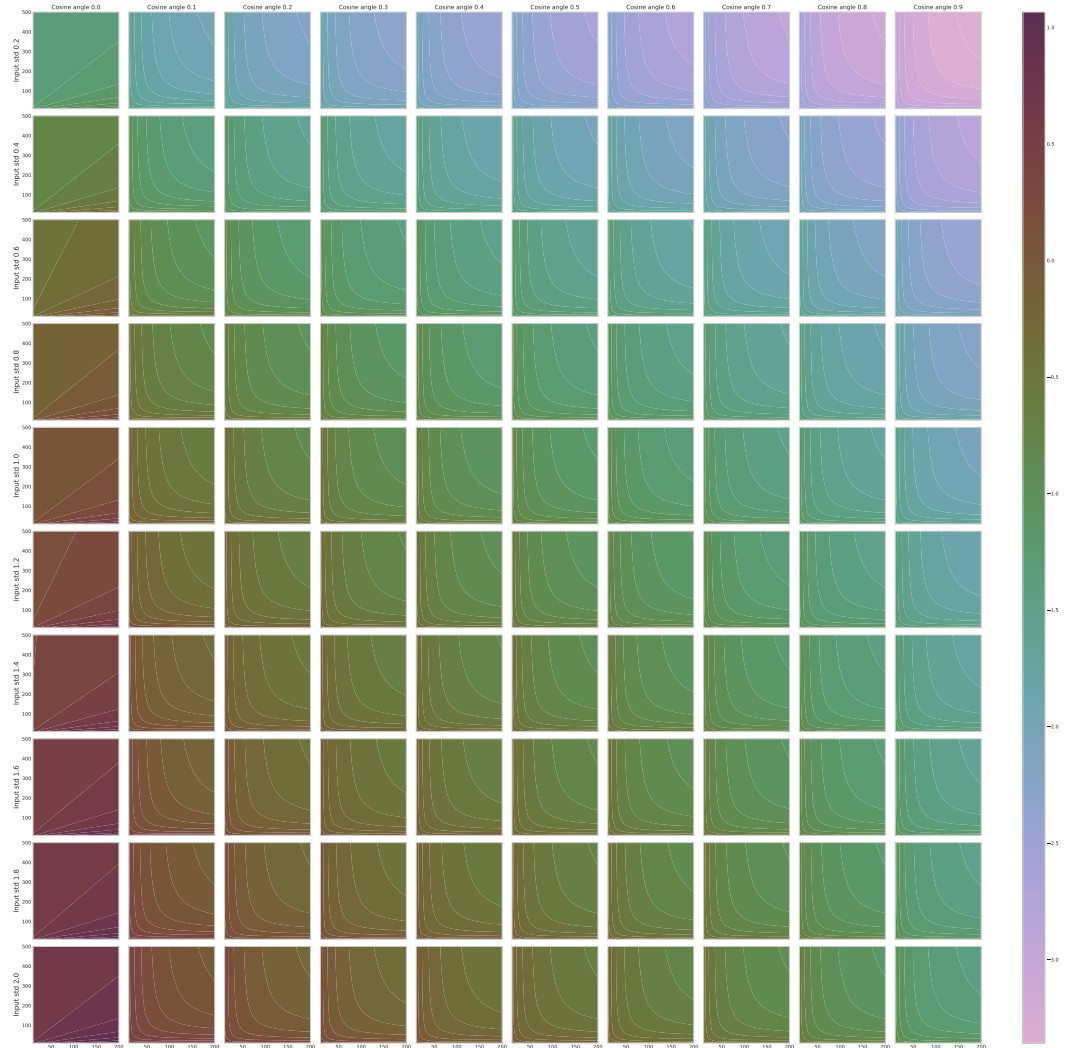

Figure 14: Log ratio of the norm of the gradients of the queries as expected by the theory, compared to those of the values for varying values of embedding dimension, sequence length, cosine of the tokens angle and standard deviation. We use Equations (17) and (18).

operation to

$$\mathbf{S}_\tau^\ell := \mathrm{softmax}\left(\frac{\tau}{\sqrt{d_k}}\mathbf{X}^\ell\mathbf{W}^Q\left(\mathbf{X}^\ell\mathbf{W}^K\right)^\top\right)\mathbf{X}^\ell\mathbf{W}^V. \tag{45}$$

By computing the gradients of the queries/keys as in 3.3, Equation 18 becomes:

$$\mathbb{E}\left\|\frac{\partial\mathbf{S}_\tau}{\partial\mathbf{W}^Q}\right\|_F^2 = \tau^2\sigma_x^6\frac{(n-1)}{n}(1-\rho)^2 d(n+d). \tag{46}$$

Hence, the norm of gradients of the queries/keys scales linearly with $\tau$. On the contrary, the temperature scaling inside the Softmax will not affect the gradients with respect to the values due to the fact that the Softmax normalizes the activations (see Lemma A.1 in Wang et al. [2022]). Hence $\tau$ can be heuristically chosen such that the magnitude of the gradients approximately matches:

$$\mathbb{E}\left\|\frac{\partial\mathbf{S}_\tau}{\partial\mathbf{W}^Q}\right\|_F^2 \overset{!}{=} \mathbb{E}\left\|\frac{\partial\mathbf{S}_\tau}{\partial\mathbf{W}^V}\right\|_F^2 \iff \tau^2 \approx \frac{dn(1+\rho(n-1))}{\sigma_x^4(1-\rho)^2(n+d)(n-1)}. \tag{47}$$

We stress that this requires a constant correlation $\rho$. In practice, this can be estimated as the mean correlation across all pairs of tokens (as we do in the computation of the correlations in Figure 1). Furthermore, both $\rho$ and the variance $\sigma_x^2$ change across layers as our analysis in Section 3.2 predicts. Hence in practice a different temperature per layer should be adopted. Finally, note that in practice both $\rho$ and $\sigma_x^2$ change during training, and is hard to study their the dynamics under SGD. We leave the time evolution of $\rho$ and $\tau$ as an exciting future direction. Also, the value of $n$ is set to be the average number of tokens per sentence. In this work, we set $\tau$ to a fixed value according to our analysis at initialization.

## C.3 Translation Task

We now describe the experimental setup regarding the translation task on the IWSLT'14 De-En dataset. Using the ideas detailed in the previous section, we choose a temperature value of $\tau_{\text{final}} = 8.5$ to match the gradient norms of the values and queries as in Equations. (17) and (18). Doing so, we assume a constant small correlation between tokens (also empirically verified in Fig. 15) and set the sequence length $n$ to the average found in our training dataset. Due to instabilities in training, we use warm-up on this temperature value. In short:

$$\tau = \tau_{\text{final}} \cdot \max(1, \frac{\text{step}}{\text{steps}_{\text{warmup}}}),$$

with 'steps$_{\text{warmup}} = 1000$' and 'step' the current training step.

We base our implementation on fairseq [Ott et al., 2019]. For the hyperparameter configuration, we mostly rely on the extensive search already done in fairseq [Ott et al., 2019] and Liu et al. [2020]. The final used parameters are exhibited in Table 3. For the final evaluation, we use the best-performing model on the left-out validation set. We apply weight decay as in Loshchilov and Hutter [2017] for both SGD and Adam.

| | Hyperparameters | Value |
|---|---|---|
| | Max tokens | 4096 |
| | Label smoothing | 0.1 |
| | clip-norm | 0.0 |
| | General Dropout | 0.3 |
| | Attention Dropout | 0.1 |
| | ReLU Dropout | 0.1 |
| | Hidden size | 512 |
| | FFN inner hidden size | 2048 |
| | Attention Heads | 4 |
| Adam | Learning rate | $7\epsilon^{-4}$ |
| | Learning rate scheduler | inverse sqrt |
| | Warm-up updates | 6000 |
| | Warm-up init learning rate | 1e-7 |
| | Adam $(\beta_1, \beta_2)$ | (0.9, 0.98) |
| | Training updates | 100K |
| | Weight decay | 0.0001 |
| SGD | Learning rate | $2\epsilon^{-2}$ |
| | Learning rate scheduler | step |
| | Step scheduler $\gamma$ | 0.1 |
| | Step scheduler update steps | [100K, 200K] |
| | Training updates | 250K |
| | Weight decay | 0.001 |

Table 3: Hyperparameters for the IWSLT'14 De-En translation task (Figure 1, 5).

Finally, in Figure 15 we display the evolution of correlations, residual scaling, and norm of the activations, with depth, for our best trained model. The residual scaling $\alpha_1, \alpha_2$ are trainable parameters. This enables them to weight differently the residual branches if deemed necessary. Although these values increase during training, the correlation between the tokens does not significantly increase, which as implied by our main results, allows efficient propagation of the gradients. The norm of the propagated forward signal tends to slightly increase with depth.

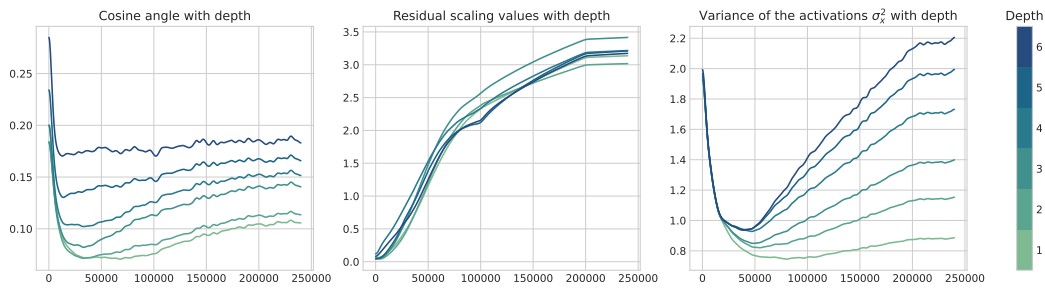

Figure 15: Evolution of the cosine of the angles, the trained residual $\alpha_1, \alpha_2$ and the activation norm throughout our training.