# OpenReview forum: "Signal Propagation in Transformers: Theoretical Perspectives and the Role of Rank Collapse"
_NeurIPS.cc/2022/Conference — NeurIPS 2022 Accept_

### Official Review · Reviewer_FzSt · 2022-07-10

**Rating:** 4
**Confidence:** 4
**Soundness:** 3 good
**Presentation:** 3 good
**Contribution:** 2 fair

**Summary:**

This work investigates the rank collapse problem for transformers at the initialization, showing that it results in vanishing gradient and slow training for self-attention parameters. Based on the theoretical study, the work proposed a method to alleviate this issue via adding a temperature parameter to the self-attention. The authors claim that the work opens the way for new, well-founded, and motivating approaches to improve optimization in these models.


**Questions:**

The reviewer’s questions relate to the weakness of this work.
1. Although the authors justify Assumption 3.1 to some extent. The paper assumes linear activation, which does not hold in practice. How does the theory well connect to the practice?
2. Some of the argument is in the setting where layers or dimensions are infinite. How do these results connect to the finite sample case?
3. The major question is about Section 3.4. It is unclear (at least not well explained) from the theory how a temperature can be helpful. And how to choose the size of the temperature parameter?


**Strengths And Weaknesses:**

*Strength: the paper is well-presented and easy to follow;
The work studies an interesting problem of the rank collapsing phenomenon at initialization.  The work provides rigorous analysis for explaining why this happens under several assumptions.
The work utilizes the understanding, showing it can be useful for more efficient training of deep networks.

*Weakness: The assumption is quite strong (Assumption 3.1), and the sigma is linear. These do not hold in general in practice deep networks. Because of the significant simplification, It is unclear about the connection between theory and practice.
To deal with the vanishing gradient problem caused by rank collapse, the author proposed to add a temperature scaling to the self-attention. It is very vague how the theory applies such a temperature would help with training. And it is unclear how to choose its size.

---

> ### Author Response · Authors · 2022-08-02
> **Part I: on the major contributions**
>
> We thank the reviewer for finding the paper well written and easy to follow, as well as for acknowledging
> the importance of our contributions on the consequences of rank collapse in Transformers. In the following,
> we address all the reviewer’s questions with the hope that we will clarify all the raised doubts.
>
>  **Major contributions of our paper**. We sincerely thank the reviewer for acknowledging the importance of our findings on the role of rank collapse in relation to vanishing gradient. We would like to add that there are additional contributions that we believe are very significant and would like to bring to the Reviewer’s attention:
> 1. As pointed out by both Reviewer 1 (Wg56) and 2 (w8ur), we show how rank collapse can be avoided by scaling the residuals with a factor $1/\sqrt{depth}$. We do so by computing **exact** formulas of the propagating tokens’ norm and pairwise correlation between different tokens.
> 2. Our exact results allow us to conclude that without the $1/\sqrt{depth}$ scaling of the residual branches, the rank quickly reaches values very close to 1, as we show in Figure 4, and formally reaches 1 in the limit $depth \to \infty$ (Theorem 3.3). This is crucial, and completes the analysis of Dong et al. [4], where it was shown that simply adding residual connections fixes the rank collapse issue. Here we show that in practice the network reaches exponentially quickly an undesirable regime of ”quasi-degeneracy”. Our analysis is exact for the linear activation case, but the issue is even stronger for ReLU nets, as we will argue below when we discuss the validity of our assumptions in more realistic settings.
> 3. In Section 3.3, we perform a careful analysis of the expected gradient norm of the queries/keys and values, finding interesting opposite trends with respect to several parameters. The results of Section 3.3 motivate the usage of the temperature scaling (Section 3.4) to fix the disproportionate gradient’s magnitude (see later for a detailed answer on this point).  We also extensively validate the theory in realistic settings (as we detail later) and have made the connection between Section 3.3 and 3.4 clearer in the revised version of the paper based on the Reviewers' comments.
>
> We hope that the reviewer agrees on these additional contributions.

---

> > ### Author Response · Authors · 2022-08-02
> > **Part II: on the assumptions**
> >
> > Our paper has two major assumptions: the "uniform-tokens" assumption and the linear activation instead of ReLU (note the latter is used only in Section 3.2). Here we discuss and bring evidence that both assumptions do not alter the main conclusions of our work in realistic settings.
> > 1. **On Assumption 3.1 and infinite width layers**. — (based on the comment *"Some of the argument
> > is in the setting where layers or dimensions are infinite. How do these results connect to the finite
> > sample case?"*) — We would like to stress that its necessity stems from the difficulty of computing
> > expectations in closed form with the softmax activation, and we provide supporting evidence and
> > relevant prior works at lines 129-131. Furthermore, Assumption 3.1 provides a faithful approximation
> > to the realistic case of the commonly-used ”Xavier” initialization for the queries and keys, as we mention
> > in lines 134-137, and we produce both theoretical and empirical supporting evidence in Appendix A5,
> > which is entirely dedicated to the investigation of the assumptions in realistic settings (in particular,
> > see Figure 7-11-12). Finally, note that Xiong et al. [1] also adopt the same assumption, but justify it differently
> > (and perhaps unrealistically) by setting the matrices $W^Q$ and $W^K$ to zero. Other papers make even
> > stronger assumptions, e.g. assume that the Softmax operates in the linear regime Wang et al. [2]. Compared to
> > these works, the motivations for our assumptions are milder and more realistic. Regarding the infinite
> > width layers, note that large enough width values behave close to the infinite case due to standard
> > concentration inequalities (which hold broadly under mild assumptions, such as finite moments). Hence,
> > **we conclude that typical values of the width in modern Transformers, together with the
> > Xavier initialization are well captured by our proposed assumption**. As a side note, the
> > infinite width assumption is commonly done in influential papers on signal propagation, as it makes
> > several problems theoretically more tractable. For instance,  Schoenholz et al. [5], Poole et al. [6] and several follow-up works adopt –
> > implicitly or explicitly – this assumption. Also, the rich literature on Neural Tangent Kernel (NTK) or
> > Neural Network Gaussian Processes (NNGP) studies the infinite width regime to understand properties
> > of neural networks.
> >
> > 2. **On the Linear activation**. — (based on the comment "*The paper assumes linear activation, which
> > does not hold in practice. How does the theory well connect to the practice?"*) — We agree with the
> > reviewer that in practice the ReLU activation is adopted. However, we would like to remark here that
> > our work mainly focuses on the issue of rank collapse, and hence how the alignment of the tokens’
> > representation increases with depth. Crucially, in this setting the linear activation function provides
> > a bound on the correlation with respect to the ReLU case. In fact, correlations are exactly preserved
> > in expectation in the linear case, but increase in the ReLU case (for instance, see the contraction
> > argument in Nachum et al. [3] below Equation (2) ). Hence, **the perfect alignment (a.k.a rank collapse) that
> > affects the linear case will (provably) affect the ReLU case as well, and the rank collapses
> > faster with depth in the ReLU case**. For an empirical verification, please see Figure 10 in the revised
> > paper, where we compare the speed of rank degeneracy for the two cases.
> > Note that the propagation of inner products under the ReLU activation does not have a closed-form
> > solution, as we detailed in the original version of this work (end of Section 3.2). Therefore, exact results
> > in the ReLU case cannot be derived. In these cases when more general statements are intractable,
> > developing a theory for simplified models is a standard practice in machine learning. We particularly
> > thank the reviewer for this very important pointer that **strengthens the significance of our work**. **We
> > have added an extensive explanation for the extension to the ReLU case in the revised
> > version of the paper** at the end of Section 3.2. Finally, note that the linear activation assumption is not present in
> > the statement of Theorem 3.2 regarding the vanishing gradient problem (which is a property
> > of the softmax) but only in the signal propagation analysis of Section 3.2.

---

> > > ### Author Response · Authors · 2022-08-02
> > > **Part III: on the connections between our theory (Section 3.3) and the temperature trick (Section 3.4)**
> > >
> > > We thank the Reviewer for their pointer on the role of the temperature $\tau$ and its connection to our theory. We have made this connection more explicit in the revised version of the paper, and added Section C.2 in the appendix to discuss the rationale behind the choice of $\tau$.
> > >
> > > 1. **Why is Tempering Helpful?** — (based on the comment "*The major question is about Section
> > > 3.4. It is unclear (at least not well explained) from the theory how a temperature can be helpful*")
> > > — Our theory (Section 3.3) derives precise expressions for the gradients of queries/keys (Eq. 18)
> > > and values (Eq. 17) under Gaussian tokens (assumption that holds at initialization in a shallow
> > > network due to the presence of the embedding layer). These equations predict opposite trends
> > > for the gradient norms w.r.t several parameters, such as the pair-wise correlation of the
> > > tokens. We verify these calculations in Figure 12 of the revised paper (already present in the
> > > original version), where theory and practice are well aligned. This difference in magnitude can be
> > > alleviated by tuning an appropriate temperature parameter $\tau$ . In fact, $\tau$ **linearly affects the
> > > norm of gradients of the queries/keys**, which becomes:
> > > $$
> > > E ||\frac{\partial S}{\partial W^Q}||_F^2 = \tau^2 \sigma_x^6 \frac{(n-1)}{n} (1 - \rho)^2 d (n + d)
> > > $$
> > > while **it only marginally affects the gradients of the values** due to the normalization
> > > applied by the Softmax (only a constant factor), as also shown in Wang et al. [2] (Lemma A.1). Hence, $\tau$ can
> > > be helpful to fix this difference in magnitude.
> > >
> > > 2. **How to choose $\tau$.** — (based on the comment "*And how to choose the size of the temperature
> > > parameter?*") — Let $S_\tau$ be the tempered Softmax. Following the intuition presented in the
> > > previous point, $\tau$ **can be chosen such that the magnitude of the gradients in Equations
> > > 17** (in its version provided in point 1 above) **and 18 are matched**:
> > > $$
> > > E || \frac{\partial S_\tau}{\partial W^Q}||_F^2 = E ||\frac{\partial S_\tau}{\partial W^V}||_F^2 \iff \tau^2 \approx \frac{dn(1 + \rho(n-1))}{\sigma_x^4(1-\rho)^2(n+d)(n-1)}.
> > > $$
> > > This formula is used to compute $\tau$ in our experiments. We provide the details for
> > > the computation of this formula in the revised manuscript in Section C.2 of the appendix, and
> > > invite the interested reviewer to assess its significance.
> > >
> > > 3. **Does the temperature scaling $\tau$ fix the vanishing gradient problem?** — (based on
> > > the comment "*To deal with the vanishing gradient problem caused by rank collapse, the author
> > > proposed to add a temperature scaling to the self-attention*") — No, it only fixes the problem of
> > > disproportionate gradient magnitude, as analyzed in Section 3.3. Under rank collapse, the tokens’
> > > correlation converges to 1 and hence one would need $\tau \to \infty$. Hence, the rank collapse problem
> > > cannot be fixed by tempering alone, but it can with the $1/\sqrt{depth}$ residual scaling (see point
> > > 2 in Part I above). We show this theoretically as well as in practical settings in
> > > Fig. 1 (e.g. top right), where the absence of residual scaling produces untrainable networks that
> > > output only random predictions. Also, in Fig. 4 (present in the original paper) and 10 (revised
> > > version) we compare the presence/absence of residual scaling in various settings, confirming our
> > > theoretical results (see Fig. 8 for extra empirical verification of our theory).
> > >
> > > 4. **Is the difference in gradients’ magnitude present in realistic Transformers?** — (based
> > > on the comment "*It is very vague how the theory applies such a temperature would help with
> > > training*") — Yes. Commonly initialized Transformers are affected. For instance, in Figure 3
> > > (b,c) — which is based on a realistic Transformer with ReLU activation — the magnitude of the
> > > gradients of the queries and keys show a drastically different trend with respect to the tokens’
> > > correlation at initialization, as our theory of Section 3.3 postulates. Also, we have shown in Fig.
> > > 6 that Adam chooses a larger learning rate for the queries and keys, indicating smaller gradients
> > > for this set of parameters. This was also independently observed for NLP tasks in Liu et al. [7] (Figure
> > > 11 of that work) — as we mention in the related work Section. Hence, we conjectured that the
> > > temperature scaling allows training without adaptive methods (Adam) because it fixes this issue
> > > of disproportionate gradient magnitudes. Our results clearly confirm it, as we are able to train
> > > a Transformer competitively using SGD (Figure 5). Acknowledging the connection between the
> > > temperature scaling and our work, we thank the reviewer for the pointer. We have made the
> > > connection more explicit in Section 3.3 and 3.4.
> > >
> > > We hope that this rebuttal together with the current version of the paper satisfies all the Reviewer’s
> > > requirements for acceptance and fully addresses all the Reviewer’s concerns. We would also be pleased
> > > to receive any further comment on this interesting discussion.

---

> > > > ### Author Response · Authors · 2022-08-02
> > > > **References**
> > > >
> > > > [1] Ruibin Xiong, Yunchang Yang, Di He, Kai Zheng, Shuxin Zheng, Chen Xing, Huishuai Zhang,
> > > > Yanyan Lan, Liwei Wang, and Tieyan Liu. On layer normalization in the transformer architecture. In
> > > > International Conference on Machine Learning, pages 10524–10533. PMLR, 2020.
> > > >
> > > > [2] Wang, Hongyu, et al. ”Deepnet: Scaling transformers to 1,000 layers.” arXiv preprint arXiv:2203.00555
> > > > (2022).
> > > >
> > > > [3] Ido Nachum, Jan H azla, Michael Gastpar, and Anatoly Khina. A johnson–lindenstrauss framework
> > > > for randomly initialized cnns. arXiv preprint arXiv:2111.02155, 2021.
> > > >
> > > > [4] Yihe Dong, Jean-Baptiste Cordonnier, and Andreas Loukas. Attention is not all you need: Pure
> > > > attention loses rank doubly exponentially with depth. In International Conference on Machine Learning,
> > > > pages 2793–2803. PMLR, 2021.
> > > >
> > > > [5] Samuel S Schoenholz, Justin Gilmer, Surya Ganguli, and Jascha Sohl-Dickstein. Deep information
> > > > propagation. arXiv preprint arXiv:1611.01232, 2016.
> > > >
> > > > [6] Ben Poole, Subhaneil Lahiri, Maithra Raghu, Jascha Sohl-Dickstein, and Surya Ganguli. Exponential
> > > > expressivity in deep neural networks through transient chaos. Advances in neural information processing
> > > > systems, 29, 2016.
> > > >
> > > > [7] Liyuan Liu, Xiaodong Liu, Jianfeng Gao, Weizhu Chen, and Jiawei Han. Understanding the difficulty of training transformers. arXiv preprint arXiv:2004.08249, 2020

---

> > > > > ### Author Response · Authors · 2022-08-08
> > > > > **Response reminder**
> > > > >
> > > > > Dear Reviewer FzSt,
> > > > >
> > > > > Due to the fact that the deadline for the author-reviewer discussion period is approaching, we are kindly reminding the reviewer to share their thoughts and suggestions about our rebuttal. Actively engaging in the discussion would definitely be of great value for us, and help us further improving our work.
> > > > >
> > > > > Thank you again for your time and efforts.

---

### Official Review · Reviewer_w8ur · 2022-07-10

**Rating:** 7
**Confidence:** 4
**Soundness:** 3 good
**Presentation:** 3 good
**Contribution:** 3 good

**Summary:**

This paper studies the rank collapse problem of the tokens' representations at initialization in Transformer. The authors provide theoretical analysis to further understand and deal with this problem. They conclude that the rank collapse problem can cause the gradient vanishing problem during training and propose that applying an appropriate depth dependent scaling of the residual branches can prevent the rank collapse problem. Besides theoretical analysis, the authors also provide some experimental results to demonstrate the effectiveness of their proposed remedy method.

**Questions:**

1. In line 31, you mentioned Dong et.al 2021 study the rank collapse problem in the absence of residual connections, I wonder whether it is proper or what the problem is for the requirement?
2. What is the Depth in Figures 3 and 4?


**Strengths And Weaknesses:**

Strengths:
1. This paper studies a novel problem, rank collapse, to further understand the optimization of Transformer. They investigate how the rank collapse hinders the training process of Transformer from the aspect of gradient vanishing, which has not yet been well studied for new architectures, such as Transformer.
2. The authors propose a remedy method, an appropriate depth dependent scaling of the residual branches, to avoid the rank collapse problem. The experimental results demonstrate the effectiveness of this approach, as shown in Table 5 and Figure 1.
3. This paper brings a new problem to studying the signal propagation in Transformers. This would motivate other researchers to further study and improve the optimization of Transformer models.
4. This paper introduces the problems, their analysis, and their solutions clearly. The paper is well written and easy to follow. The authors present their theoretical analysis clearly and easy to be understood. The notations and equations are easy to read.

Weaknesses:
1. Some theoretical analyses are based on the previous work, Dong et al. 2021. This would make the novelty of this paper a little bit trivial.
2. Section 3.4 seems to have less relationship with the main topic of this paper. To my understanding, it is redundant.
3. The authors only provide some results on NLP tasks. I wonder if this problem is still severe and if we can address it in the same way for vision transformers.

---

> ### Author Response · Authors · 2022-08-02
> **PART I**
>
> We thank Reviewer 2 (w8ur) for the thorough analysis of our work and constructive feedback. In particular, we thank the Reviewer for acknowledging the effectiveness of the proposed $1/\sqrt{depth}$ residual scaling to avoid rank collapse, as well as recognizing the importance of our results regarding connection between rank collapse and vanishing gradients. Additionally, we believe that our work contains several novel contributions, such as the analysis of the gradient norm of different parameters of the network (Section 3.3), which leads to our proposed temperature scaling of the Softmax (Section 3.4). The Reviewer is invited to read the Common Points and the answer to Reviewer 3 (FzSt) for the connection between Section 3.3 and 3.4, and for the role of the temperature inside the Softmax, respectively. Section 3.3 leads us to possible explanation on why adaptive optimization methods have been preferable with Transformers and to a remedy to boost the performance of plain stochastic gradient descent. We clarify all of the reviewer's concerns below.
>
> 1. **Novelty with respect to Dong et al. 2021**. — (based on the comment "*Some theoretical analyses
> are based on the previous work, Dong et al. 2021. This would make the novelty of this paper a little
> bit trivial.*") — Our work is indeed partly motivated by the results of Dong et al. 2021. However,
> our analysis is not based on theirs, with implications going far beyond their results. By analyzing
> the gradient propagation, we demonstrated that rank collapse leads to vanishing gradients. Crucially,
> **note that this property does not appear in regular multi-layer perceptron neural networks**
> (Daneshmand et al. 2020a) and may hinder training, as empirical evidence suggests in Figures 1,
> 5 and Table 1. Hence, **our finding highlights a pathological problem specific of Transformers** (in
> particular of the Softmax). Although the existence of residual branches is sufficient to preserve the
> rank of the representation (at finite depth) as claimed by Dong et al. 2021, inadequately scaled residual
> branches lead to a quick increase in correlations (Figure 3), which affects the parameters of the network
> differently. We then indicate how a depth-dependent scaling can fix this collapse, even for very (or even
> infinitely) deep networks. Under our analysis, we can interpret other successful initialization remedies,
> e.g. ReZero, Fixup. We then characterize precisely the discrepancy of the norm of the gradients
> between the values and queries (and keys) parameters, verifying that these hold in practice. Finally,
> we provide a possible solution to bridge the gap between adaptive methods and gradient descent, which
> directly follows from our analysis. Overall, we believe that our contributions make several steps forward
> compared to Dong et al. 2021.
>
> 2. **On the relationship between Section 3.4 and the rest of the paper**. — (based on the comment
> "*Section 3.4 seems to have less relationship with the main topic of this paper.*") — The temperature
> scaling of the Softmax helps in reducing the difference in magnitude between the gradients of the
> queries/keys and the values. We point the Reviewer to the General answer addressing all reviewers and to our answer to
> Reviewer 3 (FzSt) — in particular, Part III — for a detailed description of the relevance of section 3.4 and its connection to the
> rest of the paper.
>
> 3. **Vision Tasks**. — (based on the comment "*The authors only provide some results on NLP tasks. I
> wonder if this problem is still severe and if we can address it in the same way for vision transformers.*") — As highlighted in our general response, the insights presented are agnostic to the task but instead
> apply generally to the Transformer architecture. Producing results for NLP is standard practice. In
> our general comments we also present preliminary results on Vision Transformers, where unsurprisingly
> the same trends can be observed.

---

> > ### Author Response · Authors · 2022-08-02
> > **Part II**
> >
> > We thank the Reviewer to show significant interest in our paper. Here we reply to their curiosities regarding the work of Dong et al. 2021 and the depth in Figures 3,4.
> >
> > 1. **Question 1.** Dong et al. 2021 study the rank collapse problem in the absence of residual connections.
> > In the presence of skip connections, they hypothesize that short paths prevent pure self-attention
> > networks from degenerating to rank-1. Crucially, Dong et al. (1) did not analyze precisely the influence
> > of skip connections (just provided a path argument to support their conjectures) and (2) did not link
> > rank collapse with the gradient propagation. In our work, we show that although the existence of skip
> > connections preserves the rank, a rapid increase in the tokens correlations can lead to significantly
> > different gradient norms of the different parameters of the network. This can be clearly illustrated in
> > Figure 3, where non-scaled residual connections lead to tokens’ correlation very close to 1 with even a
> > relatively small number of stacked transformer blocks.
> >
> > 2. **Question 2**. In Figure 4, we analyze the evolution of correlations with depth, with and without the
> > use of an adequately chosen residual scaling. Depth (on the x-axis) refers to the number of transformer
> > blocks (Figure 2) stacked one after another. In practice, the commonly used activation function ReLU
> > leads to an even faster increase in correlations (for instance, see the contraction argument in Nachum et al. [1] below
> > Equation (2)). See also Figure 10 in the revised version of the paper for a comparison between using
> > linear and ReLU activation functions. In Figure 3 (left) we study the propagation of correlations with
> > depth (x-axis) for a range of different residual scaling values. The middle and right subplots in Figure
> > 3 underline the effect on the norm of the gradients for the different parameters of the network.
> >
> > We hope that our rebuttal address the Reviewer’s concerns on the significance and abundance of results
> > beyond Dong et al, 2021. Furthermore, we hope that after our improvement in the revised version, the Reviewer doubts on the connection between the theory and section 3.4 are resolved.
> >
> > [1] Ido Nachum, Jan H azla, Michael Gastpar, and Anatoly Khina. A johnson–lindenstrauss framework for randomly initialized cnns. arXiv preprint arXiv:2111.02155, 2021.

---

> > > ### Comment · Reviewer_w8ur · 2022-08-08
> > > **response to rebuttal**
> > >
> > > Thank you for the response. Most of the reviewer's concerns are addressed.

---

### Official Review · Reviewer_Wg56 · 2022-07-14

**Rating:** 5
**Confidence:** 4
**Soundness:** 3 good
**Presentation:** 3 good
**Contribution:** 3 good

**Summary:**

This paper focuses on studies about the rank collapse phenomenon of transformers regarding its causes and effects. Authors first show that rank collapse has side effect on training by causing vanishing gradients of the queries and keys of a Transformer. Then the causes of this phenomenon is described and a solution is proposed to prevent it via depth dependent scaling of the residual branches. Finally, preliminary evidences are given about high efficacy of Adam compared to SGD in training Transformers.

**Questions:**

Please refer to the weakness part in the Strengths And Weaknesses section above.

**Limitations:**

The limitations and assumptions are discussed in Section 3 of the paper.

**Strengths And Weaknesses:**

Strengths:

+ Studying the rank collapse phenomenon of transformers is an interesting topic. It is important for training depth and large Transformer models, which is the key for popular foundation models.

+ Thorough theoretical analysis are provided to describe the cause of this phenomenon and potential solution to prevent it. Assumptions and limitations are also described clearly.

+ Experimental evidences are also given to validate the theoretical analysis.


Weakness:

- Experimental evidences are mainly provided for natural language processing (NLP) area. It is not clear whether the conclusions can be generalized to computer vision tasks, e.g. the most basic image classification.

- The effectiveness of the proposed depth dependent residual branch scaling seems not linked with evaluation metrics. Does this solution help training deeper Transformer models to achieve better performance for NLP and computer vision models?

---

> ### Author Response · Authors · 2022-08-02
> **On vision tasks and the role of residual scaling.**
>
> We thank the reviewer for their useful comments and particularly for acknowledging the importance of the
> rank collapse phenomenon in Transformers, for recognizing the relevance and soundness of our theoretical
> contributions, and for appreciating our experimental evaluation. To our understanding, the main reviewer’s
> concerns were mainly centered around the lack of experiments with Vision Transformers and the supposedly
> unclear relationship between the employed evaluation metrics and the importance of residual scaling. We
> address both concerns in the following.
>
> 1. **Vision tasks**. — (based on the comment "*[...] It is not clear whether the conclusions can be generalized
> to computer vision tasks, e.g. the most basic image classification.*") — The theoretical contributions
> brought by our paper are general and not restricted to a particular type of data structure. Our
> analysis of the forward and backward passes in Transformers highlights some general – and previously
> unknown properties – of the attention mechanism with softmax activation that can serve the purpose
> of designing better architectures regardless of the specific task under consideration. The goal of our
> experimental analysis is mainly to provide support and validation to our theoretical findings and, in
> line with other recent theoretical papers on Transformers (Dong et al. [1], Wang et al. [2]), we decided to focus on well-known
> NLP tasks. Nevertheless, in order to further assess our theoretical results and to address your point,
> we present preliminary experiments with vision Transformers (see Common Points above). The results
> are in line with the NLP task reported in the paper and further confirm that Transformers can be
> trained with SGD, without layer normalization, just by controlling the magnitude of the gradients via
> a temperature parameter and by stabilizing the forward pass via appropriate residual scaling.
>
> 2. **Does the proposed residual scaling correlate with the evaluation metrics?** — (based on the
> comment "*The effectiveness of the proposed depth dependent residual branch scaling seems not linked
> with evaluation metrics.*") — The positive effects of residual scaling are rigorously described by our
> theoretical results and are also acknowledged by Reviewer 2 (w8ur) (“the authors propose a remedy
> method, an appropriate depth dependent scaling of the residual branches, to avoid the rank collapse
> problem.”). Moreover, an extensive set of experiments demonstrates the importance of residual scaling
> for successful training of Transformers, even beyond the assumptions required by our theoretical results.
> In order:
>
>     - Figure 1 shows that residual scaling is crucial to successfully train deep Transformers. Not including residual scaling indeed results in increasing the tokens’ alignment at initialization, which can in turn negatively affect optimization (training loss not converging and random predictions, see the top right plot of Figure 1). Hence, the proposed residual scaling is necessary for training a deep POST-LN Transformer. Please note that this experiment has been carried out with ReLU nonlinearities, thus revealing that our theoretical results, based on linear activations, still hold in this more realistic setting.
>
>     - Table 1 and Fig. 5: Using the proposed residual scaling plus a temperature parameter allows us to make a POST-LN Transformer (or a Transformer without LayerNorm) trained with SGD achieve superior results w.r.t to other initialization schemes. This empirically verifies the importance of the proposed scaling.
>
>     In addition, the following Figures showcase the tight connection between our theoretical results and the experiments, even beyond the introduced assumptions:
>
>     - Figure 3 shows how residual scaling stabilizes the forward and the backward passes. In particular, without residual scaling, the cosine of the angle between tokens quickly converges to 1 (Left panel) and the gradients of the queries-keys (Middle panel) rapidly vanish as the number of Transformer layers increases. The result is again obtained using a POST-LN Transformer with ReLU, thus going beyond the assumptions used to derive our theoretical results.
>
>     - Figure 4 shows how the correlation is controlled by the use of residual scaling, as predicted by our theory. In particular, without residual scaling the rank collapses exponentially fast for the POST-LN and NO-LN cases. Similar results can be obtained in the case where ReLU nonlinearities are employed. Additional results in the ReLU case have been added to the revised version of the paper (Figure 10).
>
> We hope that with the extra experiments on vision tasks, as well as the clarification on the role of residual scaling, we address the Reviewer concerns and invite them to further interactions, as well as potentially increase their score.

---

> > ### Author Response · Authors · 2022-08-02
> > **References**
> >
> > [1] Dong, Yihe, Jean-Baptiste Cordonnier, and Andreas Loukas. ”Attention is not all you need: Pure
> > attention loses rank doubly exponentially with depth.” International Conference on Machine Learning.
> > PMLR, 2021.
> >
> > [2] Wang, Hongyu, et al. ”Deepnet: Scaling transformers to 1,000 layers.” arXiv preprint arXiv:2203.00555
> > (2022).

---

> > > ### Author Response · Authors · 2022-08-08
> > > **Response reminder**
> > >
> > > Dear Reviewer Wg56,
> > >
> > > Due to the fact that the deadline for the author-reviewer discussion period is approaching, we are kindly reminding the reviewer to share their thoughts and suggestions about our rebuttal. Actively engaging in the discussion would definitely be of great value for us, and help us further improving our work.
> > >
> > > Thank you again for your time and efforts.

---

### Author Response · Authors · 2022-08-02
**General Comment**

We thank the reviewers for their valuable comments and feedback. We are glad that all the reviewers found the paper well-written and easy to follow. Furthermore, we are particularly grateful that all the reviewers agreed on the significance of our work regarding the role of rank collapse in Transformers, and acknowledged the importance of our findings on the connection between rank collapse and vanishing gradients of a subset of parameters. We also thank Reviewer 1 (Wg56) and Reviewer 2 (w8ur) for underlining the importance of our contribution on the precise analysis of the causes of rank collapse through the signal propagation analysis and the infinite depth limit of the simplified Transformer architecture. Finally, we thank the same reviewers for acknowledging the significance of the theoretically principled remedy for the rank collapse issue with the $1/\sqrt{depth}$ scaling.

 **Common Points**
- **Additional experiments on Vision Transformers and purpose of Section 3.4.** As stated in our paper, our goal is to develop a theoretical understanding of signal propagation in Transformers. **The analysis we develop is valid independently of the properties of the dataset**, and due to its theoretical and explanatory nature, the experiments have the purpose of validation, and not to improve over current state-of-the-art methods. We chose to validate our analysis on NLP datasets that are commonly used with Transformers architectures. As we will argue below, our submission makes substantial contributions, demonstrating good agreements between theory and practice. More specifically, with the experiments of Section 3.4 we highlight the impact of unbalanced gradients in the backward pass – a phenomenon that arises from our theory (Section 3.3)– and show that a trick as simple as the temperature scaling of the softmax can help in coping with it (see answer to Reviewer 3 (FzSt) for details). Finally, we conjecture that this phenomenon might explain why adaptive methods are the de-facto optimizers in Transformers. That being said, we stress that our theoretical results are general since they are derived without any prior assumptions on the type of data under consideration, hence the theory will apply to vision tasks as well. **Note that we have revised the paper by adding Section C.2 in the appendix and improving the connection between Sections 3.3 and 3.4 in the main text.**

- **Extra ViT experiment.**  To verify this, we present preliminary experiments with vision Transformers. Specifically, we use the small Vision Transformer with a patch size of 8 (ViT-S/8) for the CIFAR-10 image classification task. Results are shown in the following table. The naming convention follows the one from Table 1 in the main paper. The trend demonstrated clearly falls in line with the main conclusions of our paper; using residual scaling and an adequately chosen temperature enables a Transformer trained with SGD to match the performance of a network trained with Adam. For a detailed methodology on how to choose a value for the temperature, we point to our response to Reviewer 3. We have revised the paper to make these decisions clearer (Section C.2 in the revised version).


- We have revised the paper to include a detailed experimental setup of every figure in detail, to avoid any
misinterpretations.

 | Method | Accuracy |
 |:----------:|:----------:|
 | Adam POST-LN | 93.14% |
 | SGD POST-LN | 86.74% |
 | SGD res-scale | 92.22% |
 | SGD Temperature | 93.13% |

Table1: Image classification accuracy on CIFAR-10. The naming convention follows the one from Table 1 of the main paper. Results are averaged across 3 runs.

---

### Meta-Review · Area_Chair_ErX5 · 2022-09-12

**Recommendation:** Accept
**Confidence:** Less certain

**Metareview:**

The paper presents a theoretical analysis of "rank collapse" in transformers. The analysis is backed by several experiments on a (En-De) machine translation task. The authors do not study other application domains, but results can be expected to be highly correlated, as those experiments cover encoder-decoder architectures. The authors make and test (re-)scaling recommendations (residual scaling and a temperature parameter). While not new in themselves, those recommendations are in line with their analysis, and they allow them to train transformers to strong performance with SGD. Overall, the paper represents an increment addition to the understanding of transformer optimization and is suitable for publication.

**Award:**

No

---

### Decision · Program_Chairs · 2022-09-14

Accept